# Solving Interpretable Kernel Dimension Reduction

**Chieh Wu**, **Jared Miller**, **Yale Chang**, **Mario Sznaier**, and **Jennifer Dy**

Electrical and Computer Engineering Dept., Northeastern University, Boston, MA

## Abstract

Kernel dimensionality reduction (KDR) algorithms find a low dimensional representation of the original data by optimizing kernel dependency measures that are capable of capturing nonlinear relationships. The standard strategy is to first map the data into a high dimensional feature space using kernels prior to a projection onto a low dimensional space. While KDR methods can be easily solved by keeping the most dominant eigenvectors of the kernel matrix, its features are no longer easy to interpret. Alternatively, Interpretable KDR (IKDR) is different in that it projects onto a subspace *before* the kernel feature mapping, therefore, the projection matrix can indicate how the original features linearly combine to form the new features. Unfortunately, the IKDR objective requires a non-convex manifold optimization that is difficult to solve and can no longer be solved by eigendecomposition. Recently, an efficient iterative spectral (eigendecomposition) method (ISM) has been proposed for this objective in the context of alternative clustering. However, ISM only provides theoretical guarantees for the Gaussian kernel. This greatly constrains ISM's usage since any kernel method using ISM is now limited to a single kernel. This work extends the theoretical guarantees of ISM to an entire family of kernels, thereby empowering ISM to solve any kernel method of the same objective. In identifying this family, we prove that each kernel within the family has a surrogate $\Phi$ matrix and the optimal projection is formed by its most dominant eigenvectors. With this extension, we establish how a wide range of IKDR applications across different learning paradigms can be solved by ISM. To support reproducible results, the source code is made publicly available on `https://github.com/chieh-neu/ISM_supervised_DR`.

## 1  Introduction

The most important information for a given dataset often lies in a low dimensional space [1; 2; 3; 4; 5; 6; 7; 8; 9; 10; 11; 12; 13; 14]. Due to the ability of kernel dependence measures for capturing both linear and nonlinear relationships, they are powerful criteria for nonlinear DR [15; 16]. The standard approach is to first map the data into a high dimensional feature space prior to a projection onto a low dimensional space [17]. This approach has been preferred because it captures the nonlinear relationship with an established solution, i.e., the most dominant eigenvectors of the kernel matrix. However, since the high dimensional feature space maps the original feaues nonlinearly, it is no longer interpretable. Alternatively, if the projection onto a subspace precedes the feature mapping, the projection matrix can be obtained to inform how the original features linearly combine to form the new features. Exploiting this insight, many formulations have leveraged kernel alignment or Hilbert Schmidt Independence Criterion (HSIC) [18] to model this approach [19; 8; 16; 20; 15; 1; 21; 22; 23; 12]. Together, we refer to these approaches as Interpretable Kernel Dimension Reduction (IKDR). Unfortunately, this formulation can no longer be solved via eigendecomposition, instead, it becomes a highly non-convex manifold optimization that is computationally expensive.

Numerous approaches have been proposed to solve this complex objective. With its orthogonality constraint, it is a form of optimization on a manifold: i.e., the constraint can be modeled geometrically as a Stiefel or Grassmann manifold [24; 25; 26]. Earlier work, Boumal and Absil [27] propose to recast a similar problem on the Grassmann manifold and then apply first and second-order Riemannian trust-region methods to solve it. Theis et al. [28] employ a trust-region method for minimizing the cost function on the Stiefel manifold. Wen and Yin [29] later propose to unfold the Stiefel manifold into a flat plane and optimize on the flattened representation. While the manifold approaches perform well under smaller data sizes, they quickly become inefficient when the dimension or sample size increases, which poses a serious challenge to larger modern problems. Besides manifold approaches, Niu et al. [30] propose Dimension Growth (DG) to perform gradient descent via greedy algorithm a column at a time. By keeping the descent direction of the current column orthogonal to all previously discovered columns, DG ensures the constraint compliance.

The approaches discussed thus far have remained inefficient. Recently, Wu et al. [19] proposed the Iterative Spectral Method (ISM) for alternative clustering where their experiments on a dataset of 600 samples showed that it took DG almost 2 days while ISM finished under 2-seconds *with a lower objective cost*. Moreover, ISM retains the ability to use eigendecomposition to solve IKDR. Instead of finding the eigenvectors of kernel matrices, ISM uses a small surrogate matrix $\Phi$ to replace the kernel matrix, thereby allowing for a much faster eigendecomposition. Yet, ISM is not without its limitations. Since ISM's theoretical guarantees are specific to Gaussian kernels, repurposing ISM to other kernel methods becomes impractical, i.e., a kernel method of a single kernel significantly limits its flexibility and representational power.

In this paper, we expand ISM's theoretical guarantees to an entire family of kernels, thereby realizing ISM's potential for a wide range of applications. Within this family, each kernel is associated with a matrix $\Phi$ where its most dominant eigenvectors form the solution. Here, $\Phi$ matrices replace the concept of kernels to serve as an interchangeable component of any applicable kernel method. We further extend the family to kernels that are conic combinations of the ISM family of kernels. Here, we prove that any conic combination of kernels within the family also has an associated $\Phi$ matrix constructed using the respective conic combination of $\Phi$s.

Empowered by extending ISM's theoretical guarantees to other kernels, we present ISM as a solution to IKDR problems across several learning paradigms, including supervised DR [8; 16; 20], unsupervised DR [15; 1], semi-supervised DR[21; 22], and alternative clustering [19; 23; 30]. Indeed, we demonstrate how many of these applications can be reformulated into an identical optimization objective which ISM solves, implying a significant role for ISM that has been previously unknown.

**Our Contributions.**

- We generalize the theoretical guarantees of ISM to an entire family of kernels and propose the necessary criteria for a kernel to be included into this family.
- We generalize ISM to conic combinations of kernels from the ISM family.
- We establish that ISM can be used to solve general classes of IKDR learning paradigms.
- We present experimental evidence to highlight the generalization of ISM to a wide range of learning paradigms under a family of kernels and demonstrate its efficiency in terms of speed and better accuracies compared to competing methods.

## 2  A General Form for Interpretable Kernel Dimension Reduction

Let $X \in \mathbb{R}^{n \times d}$ be a dataset of $n$ samples with $d$ features and let $Y \in \mathbb{R}^{n \times k}$ be the corresponding labels where $k$ denotes the number of classes. Given $\kappa_X(\cdot, \cdot)$ and $\kappa_Y(\cdot, \cdot)$ as two kernel functions that applies respectively to $X$ and $Y$ to construct kernel matrices $K_X \in \mathbb{R}^{n \times n}$ and $K_Y \in \mathbb{R}^{n \times n}$. Also let $H$ be a centering matrix where $H = I - (1/n)\mathbf{1}_n \mathbf{1}_n^T$ with $H \in \mathbb{R}^{n \times n}$, $I$ as the identity matrix and $\mathbf{1}_n$ as a vector of 1s. HSIC measures the nonlinear dependence between $X$ and $Y$ whose empirical estimate is expressed as $\mathbb{H}(X, Y) = \frac{1}{(1-n)^2} \operatorname{Tr}(HK_X HK_Y)$, with $\mathbb{H}(X, Y) = 0$ denoting complete independence and $\mathbb{H}(X, Y) \gg 0$ as high dependence [18]. Additional background regarding HSIC is provided in Appendix N.

A general IKDR problem can be posed as discovering a subspace $W \in \mathbb{R}^{d \times q}$ such that $\mathbb{H}(XW, Y)$ is maximized. Since $W$ induces a reduction of dimension, we can assume that $q < d$. To prevent an unbounded solution, the subspace $W$ is constrained such that $W^T W = I$. Since this formulation

has a wide range of applications across different learning paradigms, our work investigates the commonality of these problems and discovers that various learning IKDR paradigms can be expressed as the following optimization problem:

$$\max_{W} \mathrm{Tr}(\Gamma K_{XW}) \quad \text{s.t.} \ W^T W = I, \tag{1}$$

where $\Gamma$ is a symmetric matrix commonly derived from $K_Y$. Although this objective is shared among many IKDR problems, the highly non-convex objective continues to pose a serious challenge. Therefore the realization of ISM's ability to solve Eq. (1) impacts many applications. Here, we provide several examples of this connection.

**Supervised Dimension Reduction.** In supervised DR [16; 20], both the data $X$ and the label $Y$ are known. We wish to discover a low dimensional subspace $W$ such that $XW$ is maximally dependent on $Y$ *in the nonlinear high dimensional feature space*. This problem can be cast as maximizing the HSIC between $XW$ and $Y$ where we maximize $\mathrm{Tr}(K_{XW} H K_Y H)$. Since $H K_Y H$ includes all known variables, they can be considered as a constant $\Gamma = H K_Y H$. Eq. (1) is obtained by rotating the trace terms and constraining $W$ to $W^T W = I$.

**Unsupervised Dimension Reduction.** Niu et al. [1] introduced a DR algorithm for spectral clustering based on an HSIC formulation. In unsupervised DR, we also discover a low dimensional subspace $W$ such that $XW$ is maximally dependent on $Y$. Therefore, the objective here is actually identical to the supervised objective of $\mathrm{Tr}(K_{XW} H K_Y H)$, except since $Y$ in unknown here, both $W$ and $Y$ need to be learned. By setting $K_Y = YY^T$, this problem can be solved by alternating maximization between $Y$ and $W$. When $W$ is fixed, the problem reduces down to spectral clustering [30] and $Y$ can be solved via eigendecomposition as shown in Niu et al. [1]. When $Y$ is fixed, the objective becomes the supervised formulation previously discussed.

**Semi-Supervised Dimension Reduction.** In semi-supervised DR clustering problems [22], some form of scores $\hat{Y} \in \mathbb{R}^{n \times r}$ are provided by subject experts for each sample. It is assumed that if two samples are similar, their scores should also be similar. In this case, the objective is to cluster the data given some supervised guidance from the experts. The clustering portion can be accomplished by spectral clustering [31] and HSIC can capture the supervised expert knowledge. By simultaneously maximizing the clustering quality of spectral clustering and the HSIC between the data and the expert scores, this problem is formulated as

$$\max_{W,Y} Tr(Y^T \mathcal{L}_W Y) + \mu \mathrm{Tr}(K_{XW} H K_{\hat{Y}} H), \tag{2}$$

$$\text{s.t} \quad \mathcal{L}_W = D^{-\frac{1}{2}} K_{XW} D^{-\frac{1}{2}}, W^T W = I, Y^T Y = I \tag{3}$$

where $\mu$ is a constant to balance the importance between the first and the second terms of the objective, $D \in \mathbb{R}^{n \times n}$ is the degree matrix that is a diagonal matrix with its diagonal elements defined as $D_{diag} = K_{XW} \mathbf{1}_n$. Similar to the unsupervised DR problem, this objective is solved by alternating optimization of $Y$ and $W$. Since the second term does not include $Y$, when $W$ is fixed, the objective reduces down to spectral clustering. By initializing $W$ to an identity matrix, $Y$ is initialized to the solution of spectral clustering. When $Y$ is fixed, $W$ can be solved by isolating $K_{XW}$. If we let $\Psi = H K_{\hat{Y}} H$ and $\Omega = D^{-\frac{1}{2}} YY^T D^{-\frac{1}{2}}$, maximizing Eq. (2) is equivalent to maximizing $\mathrm{Tr}[(\Omega + \mu\Psi) K_{XW}]$ subject to $W^T W = I$. At this point, it is easy to see that by setting $\Gamma = \Omega + \mu\Psi$, the problem is again equivalent to Eq. (1).

**Alternative Clustering.** In alternative clustering [30], a set of labels $\hat{Y} \in \mathbb{R}^{n \times k}$ is provided as the original clustering labels. The objective of alternative clustering is to discover an alternative set of labels that is high in clustering quality while different from the original label. In a way, this is a form of semi-supervised learning. Instead of having extra information about the clusters we desire, the supervision here indicates what we wish to avoid. Therefore, this problem can be formulated almost identically as a semi-supervised problem with

$$\max_{W,Y} \mathrm{Tr}(Y^T \mathcal{L}_W Y) - \mu \mathrm{Tr}(K_{XW} H K_{\hat{Y}} H), \tag{4}$$

$$\text{s.t} \quad \mathcal{L}_W = D^{-\frac{1}{2}} K_{XW} D^{-\frac{1}{2}}, W^T W = I, Y^T Y = I. \tag{5}$$

Given that the only difference here is a sign change before the second term, this problem can be solved identically as the semi-supervised DR problem and the sub-problem of maximizing $W$ when $Y$ is fixed can be reduced into Eq. (1).

# 3 Extending the Theoretical Guarantees to a Family of Kernels

**The ISM algorithm.** The ISM algorithm, as proposed by Wu et al. [19], solves Eq. (1) by setting the $q$ most dominant eigenvectors of a special matrix $\Phi$ as its solution $W$; we define $\Phi$ in a later section. We denote these eigenvectors in our context as $V_{\max}$ and their eigenvalues as $\Lambda$. Since $\Phi$ derived by Wu et al. [19] is a function of $W$, the new $W$ is used to construct the next $\Phi$ which we again set its $V_{\max}$ as the next $W$. This process iterates until the change in $\Lambda$ between each iteration falls below a predefined threshold $\delta$. To initialize the first $W$, the 2nd order Taylor expansion is used to approximate $\Phi$ which yields a matrix $\Phi_0$ that is independent of $W$. We supply extra detail in Appendix Q and its pseudo-code in Algorithm 1.

---
**Algorithm 1** ISM Algorithm
---
**Input :** Data $X$, kernel, Subspace Dimension $q$
**Output :** Projected subspace $W$
**Initialization :** Initialize $\Phi_0$ using Table 2.
Set $W_0$ to $V_{\max}$ of $\Phi_0$.
**while** $||\Lambda_i - \Lambda_{i-1}||_2/||\Lambda_i||_2 < \delta$ **do**
    Compute $\Phi$ using Table 3
    Set $W_k$ to $V_{\max}$ of $\Phi$
**end**

---

**Extending ISM Algorithm.** Unfortunately, the theoretical foundation of ISM is specifically tailored to the Gaussian kernel. Since the proof relies heavily on the exponential structure of the Gaussian function, extending the algorithm to other kernels seems unlikely. However, we discovered that there exists a family of kernels where each kernel possesses its own distinct pair of $\Phi/\Phi_0$ matrices. From our proof, we discovered a general formulation of $\Phi/\Phi_0$ for any kernel within the family. Moreover, since the only change is the $\Phi/\Phi_0$ pair, the ISM algorithm holds by simply substituting the appropriate $\Phi/\Phi_0$ matrices based on the kernel. We have derived several examples of $\Phi_0/\Phi$ in Tables 2 and 3 and supplied the derivation *for each kernel* in Appendices B and C.

To clarify the notations for Tables 2 and 3, given a matrix $\Psi$, we define $D_\Psi$ and $\mathcal{L}_\Psi$ respectively as the degree matrix and the Laplacian of $\Psi$ where $D_\Psi = \text{Diag}(\Psi 1_n)$ and $\mathcal{L} = D_\Psi - \Psi$. Here, Diag is a function that places the elements of a vector into the diagonal of a zero squared matrix. While $K_{XW}$ is the kernel computed from $XW$, we denote $K_{XW,p}$ as specifically a polynomial kernel of order $p$. We also denote the symbol $\odot$ as a Hadamard product between matrices.

**$\Phi$ for Common Kernels.** After deriving $\Phi/\Phi_0$ pairs for the most common kernels, we note several recurrent characteristics. First, $\Phi$ scales with the dimension $d$ instead of the size of the data $n$. Since $n \gg d$ is common across many datasets, the eigendecomposition performed on $\Phi \in \mathcal{R}^{d \times d}$ can be significantly faster while requiring less memory. Second, following Eq. (6), they are highly efficient to compute since a vectorized formulation of $\Phi$ can be derived for each kernel as shown in Table 3; commonly, they reduce to a dot product between a Laplacian matrix $\mathcal{L}$ with the data matrices $X$. The occurrence of the Laplacian is particularly surprising since nowhere in Eq. (1) suggests this relationship. Third, observe from Table 3 that $\Phi$ can be expressed as $X^T \Omega X$, where $\Omega$ is a positive semi-definite (PSD) matrix. Since the formulation of $\Phi$ without $\Omega$ is the covariance matrix $X^T X$, $\Omega$ adjusts the covariance matrix by incorporating both the kernel and the label information. In addition, by applying the Cholesky decomposition on $\Omega$ to rewrite $\Phi$ as $(X^T L)(L^T X)$, $L$ becomes a matrix that adjusts the data itself. Therefore, IKDR can be interpreted as applying PCA on the adjusted data $L^T X$ where the kernel and label information is included.

| Kernel Name | $f(\beta)$ | $a(x_i, x_j)$ | $b(x_i, x_j)$ |
|---|---|---|---|
| Linear | $\beta$ | $x_i$ | $x_j$ |
| Squared | $\beta$ | $x_i - x_j$ | $x_i - x_j$ |
| Polynomial | $(\beta + c)^p$ | $x_i$ | $x_j$ |
| Gaussian | $e^{\frac{-\beta}{2\sigma^2}}$ | $x_i - x_j$ | $x_i - x_j$ |
| Multiquadratic | $\sqrt{\beta + c^2}$ | $x_i - x_j$ | $x_i - x_j$ |

Table 1: Converting common kernels to $f(\beta)$.

| Kernel | Approximation of $\Phi$s |
|---|---|
| Linear | $\Phi_0 = X^T \Gamma X$ |
| Squared | $\Phi_0 = X^T \mathcal{L}_\Gamma X$ |
| Polynomial | $\Phi_0 = X^T \Gamma X$ |
| Gaussian | $\Phi_0 = -X^T \mathcal{L}_\Gamma X$ |
| Multiquadratic | $\Phi_0 = X^T \mathcal{L}_\Gamma X$ |

Table 2: Equations for the approximate $\Phi$s for the common kernels.

| Kernel | $\Phi$ Equations |
|---|---|
| Linear | $\Phi = X^T \Gamma X$ |
| Squared | $\Phi = X^T \mathcal{L}_\Gamma X$ |
| Polynomial | $\Phi = X^T \Psi X$ , $\Psi = \Gamma \odot K_{XW,p-1}$ |
| Gaussian | $\Phi = -X^T \mathcal{L}_\Psi X$ , $\Psi = \Gamma \odot K_{XW}$ |
| Multiquadratic | $\Phi = X^T \mathcal{L}_\Psi X$ , $\Psi = \Gamma \odot K_{XW}^{(-1)}$ |

Table 3: Equations for $\Phi$s for the common kernels.

**Extending the ISM Theoretical Guarantees.** The main theorem in Wu et al. [19] proves that a fixed point $W^*$ of Algorithm 1 is a local maximum of Eq. (1) *only if* the Gaussian kernel is used. Our work extends the theorem to a family of kernels which we refer to as the ISM family. Here, we supply the theoretical foundation for this claim by first providing the following definition.

**Definition 1.** *Given $\beta = a(x_i, x_j)^T W W^T b(x_i, x_j)$ with $a(x_i, x_j)$ and $b(x_i, x_j)$ as functions of $x_i$ and $x_j$, any twice differentiable kernel that can be written in terms of $f(\beta)$ while retaining its symmetric positive semi-definite property is an ISM kernel belonging to the ISM family with an associated $\Phi$ matrix defined as*

$$\Phi = \frac{1}{2} \sum_{i,j} \Gamma_{i,j} f'(\beta) A_{i,j}. \tag{6}$$

*where $A_{i,j} = b(x_i, x_j) a(x_i, x_j)^T + a(x_i, x_j) b(x_i, x_j)^T$.*

Since the equation for different kernels varies vastly, it is not clear how they can be reformulated into a single structure that simultaneously satisfies all ISM guarantees. Definition 1 is the key realization that unites a set of kernels into a family. Under this definition, we proved later in Theorem 1 that the Gaussian kernel within the original proof of ISM can be replaced by $f(\beta)$. Therefore, the ISM guarantees simultaneously extend to any kernel that satisfies Definition 1. As a result, a general $\Phi$ for *any ISM kernel* can be derived as shown in Eq. (6). Moreover, note that the family of potential kernels is not limited to a finite set of known kernels, instead, it extends to any conic combinations of ISM kernels. We prove in Appendix O the following proposition.

**Proposition 1.** *Any conic combination of ISM kernels is still an ISM kernel.*

**Properties of $\Phi$.** Since each ISM kernel is coupled with its own $\Phi$ matrix, $\Phi$s can conceptually replace kernels. Recall that the $V_{\max}$ of $\Phi$ *for any kernel in the ISM family* is the local maximum of Eq. (1). This central property is established in the following two theorems.

**Theorem 1.** *Given a full rank $\Phi$ with an eigengap as defined by Eq. (80) in Appendix D, a fixed point $W^*$ of Algorithm 1 satisfies the 2nd Order Necessary Conditions (Theorem 12.5 [32]) for Eq. (1) using any ISM kernel.*

**Theorem 2.** *A sequence of subspaces $\{W_k W_k^T\}_{k \in \mathbb{N}}$ generated by Algorithm 1 contains a convergent subsequence.*

Since the entire ISM proof along with its convergence guarantee is required to be revised and generalized under Definition 1, we leave the detail to Appendix D and P while presenting here only the main conclusions. Functionally, our proof is separated into two lemmas to establish $\Phi$ as a kernel surrogate. Lemma 1 concludes that given any $\Phi$ of an ISM kernel, the gradient of the Lagrangian for Eq. (1) is equivalent to $-\Phi W - W \Lambda$. Therefore, when the gradient is set to 0, the eigenvectors of $\Phi$ is equivalent to the stationary point of Eq. (1). For Lemma 2, given $\bar{\Lambda}$ as the eigenvalues associated with the eigenvectors *not chosen* and $\mathcal{C}$ as constant, it concludes that the 2nd order necessary condition is satisfied when $(\min_i \bar{\Lambda}_i - \max_j \Lambda_j) \geq \mathcal{C}$. This inequality indicates the necessity for the smallest eigenvalue among the un-chosen eigenvectors to be greater than the maximum eigenvalue of the chosen by at least $\mathcal{C}$. Therefore, given the choice of $q$ eigenvectors, the $q$ smallest eigenvalues will maximize the gap. This is equivalent to finding the most dominant eigenvectors of $\Phi$. Putting both lemmas together, we conclude that the most dominant eigenvectors of any $\Phi$ within the ISM family is the solution to Eq. (1).

**Initializing $W$ with $\Phi_0$.** After generalizing ISM, different $\Phi$s may or may not be a function of $W$. When $\Phi$ is not a function of $W$, the $V_{\max}$ of $\Phi$ is immediately the solution. However, if $\Phi$ is a

function of $W$, $\Phi$ iteratively updates from the previous $W$. This process is initialized using a $\Phi_0$ that is independent of $W$. To obtain $\Phi_0$, ISM approximates the Gaussian kernel up to the 2nd order of the Taylor series around $\beta = 0$ and discovers that the approximation of $\Phi$ is independent of $W$. Our work leverages Definition 1 and proves that a common formulation for $\Phi_0$ is possible. We formalize our finding in the following theorem and provided the proof in Appendix F.

**Theorem 3.** *For any kernel within the ISM family, a $\Phi$ independent of $W$ can be approximated with*

$$\Phi \approx \text{sign}(\nabla_\beta f(0)) \sum_{i,j} \Gamma_{i,j} A_{i,j}. \tag{7}$$

**Extending ISM to Conic Combination of Kernels.** The two lemmas of Theorem 1 highlights the conceptual convenience of working with $\Phi$ in place of kernels. This conceptual replacement extends even to conic combinations of ISM kernels. As a corollary to Theorem 1, we discovered that when a kernel is constructed through a conic combination of ISM kernels, it also has an associated $\Phi$ matrix. Remarkably, it is equivalent to the conic combination of $\Phi$s from individual kernels using the same coefficients. Formally, we propose the following corollary with its proof in Appendix M.

**Corollary 1.** *The $\Phi$ matrix associated with a conic combination of kernels is the conic combination of $\Phi$s associated with each individual kernel.*

**Complexity analysis.** Let $t$ be the number of iterations required for convergence and $n \gg d$, ISM's time complexity is dominated by the dot product between $\mathcal{L} \in \mathcal{R}^{n \times n}$ and $X \in \mathcal{R}^{n \times d}$. Together ISM has a time complexity of $O(n^2 dt)$; a significant improvement from DG $O(n^2 dq^2 t)$, or SM at $O(n^2 dqt)$. ISM is also faster since $t$ is significantly smaller. While $t$ ranges from hundreds to thousands for competing algorithms, ISM normally converges at $t < 5$. In terms of memory, ISM faces similar challenges as all kernel methods where the memory complexity is upper bounded at $O(n^2)$.

## 4 Experiments

**Datasets.** The experiment includes 5 real datasets of commonly encountered data types. Wine [33] consists of continuous data while the Cancer dataset [34] features are discrete. The Face dataset [35] is a standard dataset used for alternative clustering; it includes images of 20 people in various poses. The MNIST [36] dataset includes images of handwritten characters. The Face and the MNIST datasets are chosen to highlight ISM's ability to handle images. The Flower image by Alain Nicolas [37] is another dataset chosen for alternative clustering where we seek alternative ways to perform image segmentation. For more in-depth details on each dataset, see Appendix J.

**Experimental Setup.** We showcase ISM's efficacy on three different learning paradigms, i.e., supervised dimension reduction[20], unsupervised clustering [1], and semi-supervised alternative clustering [22]. As an optimization technique, we compare ISM in Table 4 against competing state-of-the-art manifold optimization algorithms: Dimension Growth **(DG)** [30], the Stiefel Manifold approach **(SM)** [29], and the Grassmann Manifold **(GM)** [27; 38]. To emphasize ISM family of kernels, the supervised and unsupervised results using several less conventional kernels are included in Table 5. Within this table, we also investigate using conic combination of $\Phi$s by combining the Gaussian and the polynomial kernels with center alignment [39]. Since center alignment is specific to supervised cases, this is not repeated for the unsupervised case.

For *supervised* dimension reduction, we perform SVM on $XW$ using 10-fold cross validation. For each of the 10-fold experiments, we trained $W$ and the SVM classifier only on the training set while reporting the result only on the test set, i.e., the test set was never used during the training. We repeat this process for each fold of cross-validation. From the 10-fold results in Table 4, we record the mean and the standard deviation of the run-time, cost, and accuracy. We investigate the scalability in Figure 1b by comparing the change in run-time as we increment the sample size. For *unsupervised* dimension reduction, we perform spectral clustering on $XW$ after learning $W$ where we record the run-time, cost, and NMI. For *alternative clustering*, we highlight the ISM family of kernels by reproducing the original ISM results (generated with Gaussian kernel) using the polynomial kernel. On the Flower image, each sample is a $\mathcal{R}^3$ vector. We supply the original image segmentation result as semi-supervised labels and learn an alternative way to segment the image. The original segmentation and the alternative segmentation are shown in Figure1a. For the Face dataset, each

sample is a vector vectorized from a grayscaled image of individuals. We provide the identity of individuals as the original clustering label and search for an alternative way to cluster the data.

**Evaluation Metric.** In the supervised case, the test classification accuracy from the 10-fold cross validation is recorded along with the cost and run-time. The time is broken down into days (d), hours (h), minutes (m), and seconds (s). The best results are bold for each experiment. In the unsupervised case, we report the Normalized Mutual Information (NMI) [40] to compare the clustering labels against the ground truth. For detail on how NMI is computed, see Appendix L.

**Experiment Settings.** The median of the pair-wise Euclidean distance is used as $\sigma$ for all experiments using the Gaussian kernel. Degree of 3 is used for all polynomial kernels. The dimension of subspace $q$ is set to the number of classes/clusters. The convergence threshold $\delta$ is set to 0.01. All competing algorithms use their default initialization. All datasets are centered to 0 and scaled to a standard deviation of 1. All sources are written in Python using Numpy and Sklearn [41; 42]. All experiments were conducted on Dual Intel Xeon E5-2680 v2 @ 2.80GHz, with 20 total cores. Due to limited computational resources, each run is limited to 3 days.

**Complexity Analysis of Competing Methods.** The run-time as a function of linearly increasing sample size is shown for the polynomial kernel in Figure 1b. Since the complexity analysis for ISM suggests a relationship of $O(n^2)$ with respect to the sample size, $log_2(.)$ is used for the $Y$-axis. As expected, the ISM's linear run-time growth in Figure 1b supports our analysis of $O(n^2)$ relationship. The plot for competing algorithms reported a similar linear relationship with comparable slopes. This indicates that the difference in speed is not a function of the data size, but other factors such as $q$ and $t$. Using **DG**'s complexity of $O(n^2 dq^2 t)$ as an example, it normally converges when $t$ is in the ranges of thousands. Since $q = 20$ was used in the figure, the significant speed improvement from ISM can be derived from the $q^2 t$ factor since ISM generally converges at $t$ below 5.

**Results.** Comparing against other optimization algorithms in Table 4, the results confirm ISM as a significantly faster algorithm while consistently achieving a lower cost. This disparity is especially prominent when the data dimension $q$ is higher. We highlight that for the Face dataset on the Gaussian kernel, it took DG 1.92 days, while ISM finished within 0.99 seconds: a $10^5$-fold speed difference. To further confirm these advantages, the same experiment is repeated using the *polynomial kernel* where similar results can be observed. Besides the execution time and cost, the classification accuracy across 5 datasets never falls below 95% in the supervised setting. The same datasets and techniques are repeated in an unsupervised clustering problem. While the clustering quality is comparable across the datasets, ISM clearly produces the lowest cost with the fastest execution time.

Table 5 focuses on the generalization of ISM to a family of kernels. Since Table 4 already supplied results from the Gaussian and polynomial kernel, we feature 4 more kernels to support the claim. As kernel methods treat kernels as interchangeable components of the algorithm, ISM achieves a similar effect by replacing the $\Phi$ matrix. As evidenced from the table, similar accuracy and time can be achieved with this replacement without affecting the rest of the algorithm. In many cases, the multiquadratic kernel outperforms even the Gaussian and the polynomial kernel. In a similar spirit, we repeated the same experiments in the unsupervised case and received further confirmation.

To support Corollary 1, results using a Gaussian + polynomial (G+P) kernel is also supplied in Table 5. It is not surprising that a combination of $\Phi$s is the best performing kernel. Since the union of the two kernels covers a larger feature space, the expressiveness is also greater. This result supports the claim that a conic combination of $\Phi$s can replace the same combination of kernels for Eq. (1).

To study the generalized ISM on a (semi-supervised) alternative clustering problem, we use it to recreate the results from the original paper on alternative clustering. We emphasize that our results differ in the choice of using the polynomial kernel instead of the Gaussian. From the Flower experiment, it is visually clear that the original image segmentation of 2 clusters (separated by black and white) is completely different from the alternative segmentation. For the Face data, the original clusters were grouped by the identity of the individuals while the algorithm produced 4 alternative clusters. By averaging the images of each alternative cluster, the new clustering pattern can be visually seen in Figure 1a; the samples are alternatively clustered by the pose.

By applying ISM to 3 different learning paradigms, we showcase ISM as an extremely fast optimization algorithm that can solve a wide range of IKDR problems, thereby drawing a deeper connection between these domains. Hence, the impact of generalizing ISM to other kernels is also conveniently translated to these applications.

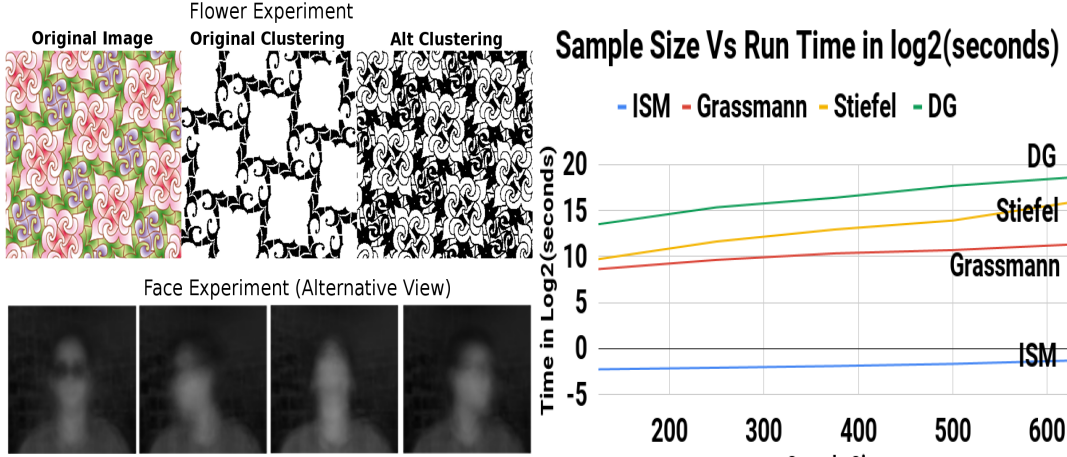

(a) Reproducing results from the original ISM paper using polynomial kernels.

(b) Log2 run-time as a function of increasing samples.

| Supervised | | Gaussian | | | | polynomial | | | |
|---|---|---|---|---|---|---|---|---|---|
| | | ISM | DG | SM | GM | ISM | DG | SM | GM |
| **Wine** | Time | **0.02s ± 0.01s** | 7.9s ± 2.9s | 1.7s ± 0.7s | 16.8m ± 3.4s | **0.02s ± 0.0s** | 13.2s ± 6.2s | 14.77s ± 0.6s | 16.82m ± 3.6s |
| | Cost | **-1311 ± 26** | -1201 ± 25 | -1310 ± 26 | -1307 ± 25 | **-114608 ± 1752** | -112440 ± 1719 | -111339 ± 1652 | -108892 ± 1590 |
| | Accuracy | **95.0% ± 5%** | 93.2% ± 5.5% | **95% ± 4.2%** | **95% ± 6%** | **97.2% ± 3.7%** | 93.8% ± 3.9% | 96.6% ± 3.7% | 96.6% ± 2.7% |
| **Cancer** | Time | **0.08s ± 0.0s** | 4.5m ± 103s | 17s ± 12s | 17.8m ± 80s | **0.13s ± 0.0s** | 4m ± 1.2m | 3.3m ± 3s | 17.5m ± 1.1m |
| | Cost | **-32249 ± 338** | -30302 ± 2297 | -31996 ± 499 | -30998 ± 560 | **-1894 ± 47** | -1882 ± 47 | -1737 ± 84 | -1690 ± 108 |
| | Accuracy | 97.3% ± 0.3% | 97.3% ± 0.3% | 97.3% ± 0.2% | **97.4% ± 0.4%** | **97.4% ± 0.3%** | 97.3% ± 0.3% | **97.4% ± 0.3%** | 97.3% ± 0.3% |
| **Face** | Time | **0.99s ± 0.1s** | 1.92d ± 11h | 10s ± 5s | 22.7m ± 18s | **0.7s ± 0.03s** | 2.1d ± 13.9h | 5.0m ± 5.7s | 21.5m ± 9.8s |
| | Cost | **-3754 ± 31** | -3431 ± 32 | -3749 ± 33 | -771 ± 28 | **-82407 ± 1670** | -78845 ± 1503 | -37907 ± 15958 | -3257 ± 517 |
| | Accuracy | **100% ± 0%** | **100% ± 0%** | **100% ± 0%** | 99.2% ± 0.2% | **100% ± 0%** | **100% ± 0%** | **100% ± 0%** | 99.8% ± 0.2% |
| **MNIST** | Time | **13.8s ± 2.3s** | > 3d | 2.5m ± 1.0s | > 3d | **12.1s ± 1.4s** | > 3d | 2.1m ± 3s | > 3d |
| | Cost | **-639 ± 2.3** | N/A | -621 ± 5.1 | N/A | **-639 ± 2** | N/A | -620 ± 5.1 | N/A |
| | Accuracy | **99% ± 0%** | N/A | 98.5% ± 0.4% | N/A | **99% ± 0%** | N/A | **99% ± 0%** | N/A |
| **Unsupervised** | | | | | | | | | |
| **Wine** | Time | **0.01s** | 9.9s | 0.6s | 16.7m | **0.02s** | 14.4s | 2.9s | 33.5m |
| | Cost | **-27.4** | -25.2 | -27.3 | -27.3 | **-1600** | -1582 | -1598 | -1496 |
| | NMI | **0.86** | **0.86** | **0.86** | **0.86** | **0.84** | **0.84** | **0.84** | 0.83 |
| **Cancer** | Time | **0.57s** | 4.3m | 3.9s | 44m | **0.5s** | 8.0m | 8.8m | 41m |
| | Cost | **-243** | -133 | -146 | -142 | **-15804** | -14094 | -15749 | -11985 |
| | NMI | **0.8** | 0.79 | **0.8** | 0.79 | 0.79 | **0.80** | 0.79 | **0.80** |
| **Face** | Time | **0.3s** | 1.3d | 5.3s | 55.9m | **1.0s** | > 3d | 22m | 1.6d |
| | Cost | **-169.3** | -167.7 | -168.9 | -37 | **-368** | NA | -348 | -321 |
| | NMI | 0.94 | **0.95** | 0.93 | 0.89 | **0.94** | N/A | 0.89 | 0.89 |
| **MNIST** | Time | **1.8h** | > 3d | 1.3d | > 3d | **8.3m** | > 3d | 0.9d | > 3d |
| | Cost | **-2105** | N/A | -2001 | N/A | **-51358** | N/A | -51129 | N/A |
| | NMI | **0.47** | N/A | 0.46 | N/A | **0.32** | N/A | **0.32** | N/A |

Table 4: Run-time, cost, and objective performance are recorded under supervised/unsupervised objectives. ISM is significantly faster compared to other optimization techniques while achieving lower objective cost.

| | | Supervised | | | | | Unsupervised | | |
|---|---|---|---|---|---|---|---|---|---|
| | | Linear | Squared | Multiquad | G+P | | Linear | Squared | Multiquad |
| **Wine** | Time | **0.003s ± 0s** | 0.01s ± 0s | 0.02s ± 0.01s | 0.007s ± 0s | Time | **0.02s** | 0.04s | 0.06s |
| | Accuracy | 97.2% ± 2.8% | 96.6% ± 3.7% | 97.2% ± 3.7% | **98.3% ± 2.6%** | NMI | 0.85 | 0.85 | **0.88** |
| **Cancer** | Time | **0.02s ± 0.002s** | 0.09s ± 0.02s | 0.15s ± 0.01s | 0.06s ± 0.004s | Time | **0.23s** | 0.5s | 0.56s |
| | Accuracy | 97.2% ± 0.3% | 97.3% ± 0.04% | **97.4% ± 0.003%** | **97.4% ± 0.003%** | NMI | 0.80 | 0.79 | **0.84** |
| **Face** | Time | **0.2s ± 0.2s** | 0.3s ± 0.2s | 0.3s ± 0.2s | 0.5s ± 0.03s | Time | **0.68s** | 0.92s | 3.7s |
| | Accuracy | 97.3% ± 0.3% | 97.1% ± 0.4% | 97.3% ± 0.4% | **100% ± 0%** | NMI | 0.93 | **0.95** | 0.92 |
| **MNIST** | Time | **6.4s ± 0.4s** | 17.4s ± 0.4s | 10.6m ± 1.9m | 17.6s ± 2.5s | Time | **3.1m** | 4.7m | 52m |
| | Accuracy | 99.1% ± 0.1% | **99.3% ± 0.2%** | 99.1% ± 0.1% | **99.3% ± 0.2%** | NMI | **0.54** | **0.54** | **0.54** |

Table 5: Run-time and objective performance are recorded across several kernels within the ISM family. It confirms the usage of $\Phi$ or linear combination of $\Phi$ in place of kernels.

# 5 Conclusion

We have extended the theoretical guarantees of ISM to a family of kernels beyond the Gaussian kernel via the discovery of the $\Phi$ matrix. Our theoretical analysis proves that the family of ISM kernels extend even to conic combinations of ISM kernels. With this extension, ISM becomes an efficient solution for a wide range of supervised, unsupervised and semi-supervised applications. Our experimental results confirm the efficiency of the algorithm while showcasing its wide impact across many domains.

**Acknowledgments**

We would like to acknowledge support for this project from NSF grant IIS-1546428. We would also like to thank Zulqarnain Khan for his insightful discussions.

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
