[Supplementary Material · ISM_supplementary.pdf]

# Appendix A    Kernel Definitions

Here we provide the definition of each kernel with relation to the projection matrix $W$ in terms of the kernel and as a function of $\beta = \mathbf{a}WW^T\mathbf{b}$.

**Linear Kernel**

$$k(x_i, x_j) = x_i^T WW^T x_j, \qquad f(\beta) = \beta. \tag{8}$$

**Polynomial Kernel**

$$k(x_i, x_j) = (x_i^T WW^T x_j + c)^p, \qquad f(\beta) = (\beta + c)^p. \tag{9}$$

**Gaussian Kernel**

$$k(x_i, x_j) = e^{-\frac{(x_i - x_j)^T WW^T (x_i - x_j)}{2\sigma^2}}, \qquad f(\beta) = e^{-\frac{\beta}{2\sigma^2}}. \tag{10}$$

**Squared Kernel**

$$k(x_i, x_j) = (x_i - x_j)^T WW^T (x_i - x_j), \qquad f(\beta) = \beta. \tag{11}$$

**Multiquadratic Kernel**

$$k(x_i, x_j) = \sqrt{(x_i - x_j)^T WW^T (x_i - x_j) + c^2}, \qquad f(\beta) = \sqrt{\beta + c^2}. \tag{12}$$

# Appendix B    Derivation for each $\Phi_0$

Using Eq. (106), we know that

$$\Phi_0 = \text{sign}(\mu) \sum_{i,j} \Gamma_{i,j} A_{i,j}. \tag{13}$$

If $\mathbf{a}$ and $\mathbf{b}$ are both defined as $x_i - x_j$, then

$$\Phi_0 = \text{sign}(4\mu) X^T (D_\Gamma - \Gamma) X. \tag{14}$$

However, if $\mathbf{a}$ and $\mathbf{b}$ are defined as $(x_i, x_j)$, then

$$\Phi_0 = \text{sign}(2\mu) X^T \Gamma X. \tag{15}$$

Therefore, to compute $\Phi_0$, the key is to first determine the $(\mathbf{a}, \mathbf{b})$ based on the kernel and then find $\mu$ to determine the sign.

**$\Phi_0$ for the Linear Kernel:** With a Linear Kernel, $(\mathbf{a}, \mathbf{b})$ uses $(x_i, x_j)$, therefore Eq. (15) is use. Since $f(\beta) = \beta$, the sign of the gradient with respect to $\beta$ is

$$\text{sign}(2\nabla_\beta f(\beta)) = \text{sign}(2) = 1. \tag{16}$$

Therefore,

$$\Phi_0 = X^T \Gamma X. \tag{17}$$

**$\Phi_0$ for the Polynomial Kernel:** With a Polynomial Kernel, $(\mathbf{a}, \mathbf{b})$ uses $(x_i, x_j)$, therefore Eq. (15) is use. Since $f(\beta) = (\beta + c)^p$, the sign of the gradient with respect to $\beta$ is

$$\text{sign}(2\nabla_\beta f(\beta)) = \text{sign}(2p(\beta + c)^{p-1}) = 1. \tag{18}$$

Therefore,

$$\Phi_0 = X^T \Gamma X. \tag{19}$$

**$\Phi_0$ for the Gaussian Kernel:** With a Gaussian Kernel, $(\mathbf{a}, \mathbf{b})$ uses $x_i - x_j$, therefore Eq. (14) is use. Since $f(\beta) = e^{-\frac{\beta}{2\sigma^2}}$, the sign of the gradient with respect to $\beta$ is

$$\text{sign}(4\nabla_\beta f(\beta)) = \text{sign}(-\frac{4}{2\sigma^2} e^{-\frac{\beta}{2\sigma^2}}) = -1. \tag{20}$$

Therefore,

$$\Phi_0 = -X^T (D_\Gamma - \Gamma) X. \tag{21}$$

**$\Phi_0$ for the RBF Relative Kernel:** With a RBF Relative Kernel, it is easier to start with the Lagrangian once we have approximated relative Kernel with the 2nd order Taylor expansion as

$$\mathcal{L} \approx -\sum_{i,j} \Gamma_{i,j} \left[ 1 + \mathrm{Tr}(W^T (-\frac{1}{\sigma_i \sigma_j} A_{i,j}) W) \right] - \mathrm{Tr} \left[ \Lambda (W^T W - I) \right]. \tag{22}$$

The gradient of the Lagrangian is therefore

$$\nabla_W \mathcal{L} \approx \left[ \sum_{i,j} \Gamma_{i,j} (\frac{2}{\sigma_i \sigma_j} A_{i,j}) \right] W - 2W\Lambda. \tag{23}$$

Setting the gradient to 0, we get

$$\left[ \sum_{i,j} (\frac{1}{\sigma_i \sigma_j} \Gamma_{i,j} A_{i,j}) \right] W = W\Lambda. \tag{24}$$

If we let $\Sigma_{i,j} = \frac{1}{\sigma_i \sigma_j}$ and $\Psi = \Sigma \odot \Gamma$, then we end up with

$$4 \left[ X^T (D_\Psi - \Psi) X \right] W = W\Lambda. \tag{25}$$

This equation requires $W$ to be the eigenvectors associated with the smallest eigenvalues. We flip the sign so the most dominant eigenvectors are the solution. Therefore, we define $\Phi$ as

$$\Phi = -X^T (D_\Psi - \Psi) X \tag{26}$$

**$\Phi_0$ for the Squared Kernel:** With a Squared Kernel, $(\mathbf{a}, \mathbf{b})$ uses $x_i - x_j$, therefore Eq. (14) is use. Since $f(\beta) = \beta$, the sign of the gradient with respect to $\beta$ is

$$\mathrm{sign}(4\nabla_\beta f(\beta)) = \mathrm{sign}(4) = 1. \tag{27}$$

Therefore,

$$\Phi_0 = X^T (D_\Gamma - \Gamma) X. \tag{28}$$

**$\Phi_0$ for the Multiquadratic Kernel:** With a Multiquadratic Kernel, $(\mathbf{a}, \mathbf{b})$ uses $x_i - x_j$, therefore Eq. (14) is use. Since $f(\beta) = \sqrt{\beta + c^2}$, the sign of the gradient with respect to $\beta$ is

$$\mathrm{sign}(4\nabla_\beta f(\beta)) = \mathrm{sign}(\frac{4}{2}(\beta + c^2)^{-1/2}) = 1. \tag{29}$$

Therefore,

$$\Phi_0 = X^T (D_\Gamma - \Gamma) X. \tag{30}$$

## Appendix C    Derivation for each $\Phi$

Using Eq. (6), we know that

$$\Phi = \frac{1}{2} \sum_{i,j} \Gamma_{i,j} [\nabla_\beta f(\beta)] A_{i,j}. \tag{31}$$

If we let $\Psi = \Gamma_{i,j} [\nabla_\beta f(\beta)]$ then $\Phi$ can also be written as

$$\Phi = \frac{1}{2} \sum_{i,j} \Psi_{i,j} A_{i,j}. \tag{32}$$

If $\mathbf{a}$ and $\mathbf{b}$ are both defined as $x_i - x_j$, then

$$\Phi = 2X^T (D_\Psi - \Psi) X. \tag{33}$$

However, if $\mathbf{a}$ and $\mathbf{b}$ are defined as $(x_i, x_j)$, then

$$\Phi = X^T \Psi X. \tag{34}$$

Therefore, to compute $\Phi$, the key is to first determine the $(\mathbf{a}, \mathbf{b})$ based on the kernel and then find the appropriate $\Psi$.

**$\Phi$ for the Linear Kernel:** With a Linear Kernel, $(\mathbf{a}, \mathbf{b})$ uses $(x_i, x_j)$, therefore Eq. (34) is use. Since $f(\beta) = \beta$, $\Phi$ becomes

$$\Phi = \frac{1}{2}\sum_{i,j}\Gamma_{i,j}[\nabla_\beta f(\beta)]A_{i,j} = \frac{1}{2}\sum_{i,j}\Gamma_{i,j}A_{i,j}. \tag{35}$$

Since, we are only interested in the eigenvectors of $\Phi$ only the sign of the constants are necessary. Therefore,

$$\Phi = \text{sign}(1)X^T\Gamma X = X^T\Gamma X. \tag{36}$$

**$\Phi$ for the Polynomial Kernel:** With a Polynomial Kernel, $(\mathbf{a}, \mathbf{b})$ uses $(x_i, x_j)$, therefore Eq. (34) is use. Since $f(\beta) = (\beta + c)^p$, $\Phi$ becomes

$$\Phi = \frac{1}{2}\sum_{i,j}\Gamma_{i,j}[\nabla_\beta f(\beta)]A_{i,j} = \frac{1}{2}\sum_{i,j}\Gamma_{i,j}[p(\beta + c)^{p-1}]A_{i,j}. \tag{37}$$

Since $p$ is a constant, and $K_{XW,p-1} = (\beta + c)^{p-1}$ is the polynomial kernel itself with power of $(p-1)$, $\Psi$ becomes

$$\Psi = \Gamma \odot K_{XW,p-1}, \tag{38}$$

and

$$\Phi = \text{sign}(p)X^T\Psi X = X^T\Psi X \tag{39}$$

**$\Phi$ for the Gaussian Kernel:** With a Gaussian Kernel, $(\mathbf{a}, \mathbf{b})$ uses $x_i - x_j$, therefore Eq. (14) is use. Since $f(\beta) = e^{-\frac{\beta}{2\sigma^2}}$, $\Phi$ becomes

$$\Phi = \frac{1}{2}\sum_{i,j}\Gamma_{i,j}[\nabla_\beta f(\beta)]A_{i,j} = \frac{1}{2}\sum_{i,j}\Gamma_{i,j}[-\frac{1}{2\sigma^2}e^{-\frac{\beta}{2\sigma^2}}]A_{i,j} = -\frac{1}{4\sigma^2}\sum_{i,j}\Gamma_{i,j}[K_{XW}]_{i,j}A_{i,j}. \tag{40}$$

If we let $\Psi = \Gamma \odot K_{XW}$, then

$$\Phi = \text{sign}(-\frac{2}{4\sigma^2})X^T(D_\Psi - \Psi)X = -X^T(D_\Psi - \Psi)X. \tag{41}$$

**$\Phi$ for the Squared Kernel:** With a Squared Kernel, $(\mathbf{a}, \mathbf{b})$ uses $x_i - x_j$, therefore Eq. (14) is use. Since $f(\beta) = \beta$, $\Phi$ becomes

$$\Phi = \frac{1}{2}\sum_{i,j}\Gamma_{i,j}[\nabla_\beta f(\beta)]A_{i,j} = \frac{1}{2}\sum_{i,j}\Gamma_{i,j}A_{i,j}. \tag{42}$$

Therefore,

$$\Phi = \text{sign}(1)X^T(D_\Gamma - \Gamma)X = X^T(D_\Gamma - \Gamma)X. \tag{43}$$

**$\Phi$ for the Multiquadratic Kernel:** With a Multiquadratic Kernel, $(\mathbf{a}, \mathbf{b})$ uses $x_i - x_j$, therefore Eq. (14) is use. Since $f(\beta) = \sqrt{\beta + c^2}$, $\Phi$ becomes

$$\Phi = \frac{1}{2}\sum_{i,j}\Gamma_{i,j}[\nabla_\beta f(\beta)]A_{i,j} = \frac{1}{2}\sum_{i,j}\Gamma_{i,j}[\frac{1}{2}(\beta + c^2)^{-1/2}]A_{i,j} = \frac{1}{4}\sum_{i,j}\Gamma_{i,j}[K_{XW}]_{i,j}^{(-1)}A_{i,j}. \tag{44}$$

If we let $\Psi = \Gamma \odot K_{XW}^{(-1)}$, then

$$\Phi = \text{sign}(\frac{1}{4})X^T(D_\Psi - \Psi)X = X^T(D_\Psi - \Psi)X. \tag{45}$$

**$\Phi_0$ for the RBF Relative Kernel:** With a RBF Relative Kernel, we start with the initial Lagrangian

$$\mathcal{L} = \sum_{i,j}\Gamma_{i,j}e^{-\frac{Tr(W^T A_{i,j} W)}{2\sigma_i\sigma_j}} - \text{Tr}(\Lambda(W^TW - I)) \tag{46}$$

where the gradient becomes

$$\nabla_W \mathcal{L} = -\sum_{i,j} \frac{1}{\sigma_i \sigma_j} \Gamma_{i,j} e^{-\frac{Tr(W^T A_{i,j} W)}{2\sigma_i \sigma_j}} A_{i,j} W - 2W\Lambda. \tag{47}$$

If we let $\Sigma_{i,j} = \frac{1}{\sigma_i \sigma_j}$ then we get

$$\nabla_W \mathcal{L} = -\sum_{i,j} \Psi_{i,j} A_{i,j} W - 2W\Lambda, \tag{48}$$

where $\Psi_{i,j} = \Sigma_{i,j} \Gamma_{i,j} K_{XW_{i,j}}$. If we apply Appendix I and set the gradient to 0, then we get

$$-4 \left[ X^T (D_\Psi - \Psi) X \right] W = 2W\Lambda. \tag{49}$$

From here, we see that it has the same form as the Gaussian kernel, with $\Psi$ defined as $\Psi = \Sigma \odot \Gamma \odot K_{XW}$.

This equation requires $W$ to be the eigenvectors associated with the smallest eigenvalues. We flip the sign so the most dominant eigenvectors are the solution. Therefore, we define $\Phi$ as

$$\Phi = X^T (D_\Psi - \Psi) X \tag{50}$$

## Appendix D    Proof for Theorem 1

The main body of the proof is organized into two lemmas where the 1st lemma will prove the 1st order condition and the 2nd lemma will prove the 2nd order condition. For convenience, we included the 2nd Order Necessary Condition [32] in Appendix G. We also convert the optimization problem into a standard minimization form where we solve

$$\min_W - \text{Tr}(\Gamma K_{XW}) \quad \text{s.t.} \ W^T W = I. \tag{51}$$

The proof is initialized by manipulating the different kernels into a common form. If we let $\beta = a(x_i, x_j) W W^T b(x_i, x_j)$, then the kernels in this family can be expressed as $f(\beta)$. This common form allows a universal proof that works for all kernels that belongs to the ISM family. Depending on the kernel, the definition of $f$, $a(x_i, x_j)$ and $b(x_i, x_j)$ are listed in Table 6. Kernels in this form are functions of the Grassmannian $WW^T$.

| Name | $f(\beta)$ | $a(x_i, x_j)$ | $b(x_i, x_j)$ |
|---|---|---|---|
| Linear | $\beta$ | $x_i$ | $x_j$ |
| Polynomial | $(\beta + c)^p$ | $x_i$ | $x_j$ |
| Gaussian | $e^{\frac{-\beta}{2\sigma^2}}$ | $x_i - x_j$ | $x_i - x_j$ |
| Squared | $\beta$ | $x_i - x_j$ | $x_i - x_j$ |

Table 6: Common components of different Kernels.

**Lemma 1.** *Given $\mathcal{L}$ as the Lagrangian of Eq. (1), if $W^*$ is a fixed point of Algorithm 1, and $\Lambda^*$ is a diagonal matrix of its corresponding eigenvalues, then*

$$\nabla_W \mathcal{L}(W^*, \Lambda^*) = 0, \tag{52}$$
$$\nabla_\Lambda \mathcal{L}(W^*, \Lambda^*) = 0. \tag{53}$$

*Proof.* Since $\text{Tr}(\Gamma K_{XW}) = \sum_{i,j} \Gamma_{i,j} K_{XW_{i,j}}$, where the subscript indicates the $i, j$th element of the associated matrix. If we let $\mathbf{a} = a(x_i, x_j), \mathbf{b} = b(x_i, x_j)$, the Lagrangian of Eq. (1) becomes

$$\mathcal{L}(W, \Lambda) = -\sum_{ij} \Gamma_{ij} f(\mathbf{a}^T W W^T \mathbf{b}) - \text{Tr}[\Lambda(W^T W - I)]. \tag{54}$$

The gradient of the Lagrangian with respect to $W$ is

$$\nabla_W \mathcal{L}(W, \Lambda) = -\sum_{ij} \Gamma_{ij} f'(\mathbf{a}^T W W^T \mathbf{b})(\mathbf{b}\mathbf{a}^T + \mathbf{a}\mathbf{b}^T)W - 2W\Lambda. \tag{55}$$

If we let $A_{i,j} = \mathbf{b}\mathbf{a}^T + \mathbf{a}\mathbf{b}^T$ then setting $\nabla_W \mathcal{L}(W, \Lambda)$ of Eq. (55) to 0 yields the relationship

$$0 = \left[ -\frac{1}{2} \sum_{ij} \Gamma_{ij} f'(\mathbf{a}^T WW^T \mathbf{b}) A_{i,j} \right] W - W\Lambda. \tag{56}$$

Since $f'(\mathbf{a}^T WW^T \mathbf{b})$ is a scalar value that depends on indices $i, j$, we multiply it by $-\frac{1}{2}\Gamma_{i,j}$ to form a new variable $\Psi_{i,j}$. Then Eq. (56) can be rewritten as

$$\left[ \sum_{ij} \Psi_{ij} A_{i,j} \right] W = W\Lambda. \tag{57}$$

To match the form shown in Table 2, Appendix H further showed that if $\mathbf{a}$ and $\mathbf{b}$ is equal to $x_i$ and $x_j$, then

$$\left[ \sum_{ij} \Psi_{ij} A_{i,j} \right] = 2X^T \Psi X. \tag{58}$$

From Appendix I, if $\mathbf{a}$ and $\mathbf{b}$ are equal to $x_i - x_j$, then

$$\left[ \sum_{ij} \Psi_{ij} A_{i,j} \right] = 4X^T [D_\Psi - \Psi] X. \tag{59}$$

If we let $\Phi = \left[ \sum_{ij} \Psi_{ij} A_{i,j} \right]$, it yields the relationship $\Phi W = W\Lambda$ where the eigenvectors of $\Phi$ satisfies the 1st order condition of $\nabla_W \mathcal{L}(W^*, \Lambda^*) = 0$. The gradient with respect to $\Lambda$ yields the expected constraint

$$\nabla_\Lambda \mathcal{L} = W^T W - I. \tag{60}$$

Since the eigenvectors of $\Phi$ is orthonormal, the condition $\nabla_\Lambda \mathcal{L} = 0 = W^T W - I$ is also satisfied. Observing these 2 properties, Lemma 1 confirms that the eigenvectors of $\Phi$ also satisfies the 1st order condition from Eq. (1).

$\square$

**Lemma 2.** *Given a full rank $\Phi$, an eigengap defined by Eq. (80), and $W^*$ as the fixed point of Algorithm 1, then*

$$\begin{aligned} \mathrm{Tr}(Z^T \nabla_{WW}^2 \mathcal{L}(W^*, \Lambda^*) Z) \geq 0 \\ \text{for all } Z \neq 0, \text{with } \nabla h(W^*)^T Z = 0. \end{aligned} \tag{61}$$

*Proof.* To proof Lemma 2, we must relate the concept of eigengap to the conditions of

$$\mathrm{Tr}(Z^T \nabla_{WW}^2 \mathcal{L}(W^*, \Lambda^*) Z) \geq 0 \quad \forall \quad Z \neq 0 \quad \text{with} \quad \nabla h(W^*)^T Z = 0 \ . \tag{62}$$

Given the constraint $h(W) = W^T W - I$, we start by computing the constrain $\nabla h(W^*)^T Z = 0$. Given

$$\nabla h(W^*)^T Z = \lim_{t \to 0} \frac{\partial}{\partial t} h(W + tZ), \tag{63}$$

the constraint becomes

$$\begin{aligned} \nabla h(W^*)^T Z = 0 &= \lim_{t \to 0} \frac{\partial}{\partial t}[(W + tZ)^T (W + tZ) - I], \\ 0 &= \lim_{t \to 0} \frac{\partial}{\partial t}[(W^T W + t W^T Z + t Z^T W + t^2 Z^T Z) - I], \\ 0 &= \lim_{t \to 0} W^T Z + Z^T W + 2t Z^T Z. \end{aligned} \tag{64}$$

By setting the limit to 0, an important relationship emerges as

$$0 = W^T Z + Z^T W. \tag{65}$$

Given a full rank operator $\Phi$, its eigenvectors must span the complete $\mathcal{R}^d$ space. If we let $W$ and $\bar{W}$ represent the eigenvectors chosen and not chosen respectively from Algorithm 1, and let $B$ and $\bar{B}$ be scambling matrices, then the matrix $Z \in \mathcal{R}^{d \times q}$ can be rewritten as

$$Z = WB + \bar{W}\bar{B}. \tag{66}$$

It should be noted that since $W$ and $\bar{W}$ are eigenvalues of the symmetric matrix $\Phi$, they are orthogonal to each other, i.e., $W^T \bar{W} = 0$. Furthermore, if we replace $Z$ in Eq. (65) with Eq. (66), we get the condition

$$\begin{aligned} 0 &= W^T(WB + \bar{W}\bar{B}) + (WB + \bar{W}\bar{B})^T W \\ 0 &= B + B^T. \end{aligned} \tag{67}$$

From Eq. (67), we observe that $B$ must be a antisymmetric matrix because $B = -B^T$. Next, we work to compute the inequality of of Eq. (62) by noting that

$$\nabla^2_{WW}\mathcal{L}(W, \Lambda)Z = \lim_{t \to 0} \frac{\partial}{\partial t} \nabla \mathcal{L}(W + tZ). \tag{68}$$

Also note that Lemma 1 has already computed $\nabla_W \mathcal{L}(W)$ as

$$\nabla_W \mathcal{L}(W) = -\frac{1}{2} \left[ \sum_{i,j} \Gamma_{i,j} f'(\beta) A_{i,j} \right] W - W\Lambda. \tag{69}$$

Since we need $\nabla_W \mathcal{L}$ to be a function of $W + tZ$ with $t$ as the variable, we change $\beta(W)$ into $\beta(W + tZ)$ with

$$\begin{aligned} \beta(W + tZ) &= \boldsymbol{a}(W + tZ)(W + tZ)^T \boldsymbol{b}, \\ &= \boldsymbol{a}^T WW^T \boldsymbol{b} + [\boldsymbol{a}^T(WZ^T + ZW^T)\boldsymbol{b}]t + [\boldsymbol{a}^T ZZ^T \boldsymbol{b}]t^2, \\ &= \beta + c_1 t + c_2 t^2, \end{aligned} \tag{70}$$

where $\beta$, $c_1$, and $c_2$ are constants with respect to $t$. Using the $\beta$ from Eq. (70) with $\nabla_W \mathcal{L}$, we get

$$\nabla^2_{WW}\mathcal{L}(W, \Lambda)Z = \lim_{t \to 0} \frac{\partial}{\partial t} \left[ -\frac{1}{2} \sum_{i,j} \Gamma_{i,j} f'(\beta + c_1 t + c_2 t^2) A_{i,j} \right] (W + tZ) - (W + tZ)\Lambda. \tag{71}$$

If we take the derivative with respect to $t$ and then set the limit to 0, we get

$$\nabla^2_{WW}\mathcal{L}(W, \Lambda)Z = \left[ -\frac{1}{2} \sum_{i,j} \Gamma_{i,j} f''(\beta) c_1 A_{i,j} \right] W + \left[ -\frac{1}{2} \sum_{i,j} \Gamma_{i,j} f'(\beta) A_{i,j} \right] Z - Z\Lambda. \tag{72}$$

Next, we notice the definition of $\Phi = -\frac{1}{2} \sum \Gamma_{i,j} f'(\beta) A_{i,j}$ from Lemma 1, the term $\mathrm{Tr}(Z^T \nabla^2_{WW}\mathcal{L}(W, \Lambda)Z)$ can now be expressed as 3 separate terms as

$$\mathrm{Tr}(Z^T \nabla^2_{WW}\mathcal{L}(W, \Lambda)Z) = \mathcal{T}_1 + \mathcal{T}_2 + \mathcal{T}_3, \tag{73}$$

where

$$\mathcal{T}_1 = \mathrm{Tr}\left( Z^T \left[ -\frac{1}{2} \sum_{i,j} \Gamma_{i,j} f''(\beta) c_1 A_{i,j} \right] W \right), \tag{74}$$

$$\mathcal{T}_2 = \mathrm{Tr}(Z^T \Phi Z), \tag{75}$$

$$\mathcal{T}_3 = -\mathrm{Tr}(Z^T Z \Lambda). \tag{76}$$

Since $\mathcal{T}_1$ cannot be further simplified, the concentration will be on $\mathcal{T}_2$ and $\mathcal{T}_3$. If we let $\bar{\Lambda}$ and $\Lambda$ be the corresponding eigenvlaue matrices associated with $\bar{W}$ and $W$, by replacing $Z$ in $\mathcal{T}_2$ from Eq. (75), we get

$$\begin{aligned} \mathrm{Tr}(Z^T \Phi Z) &= \mathrm{Tr}((WB + \bar{W}\bar{B})^T \Phi (WB + \bar{W}\bar{B})) \\ &= \mathrm{Tr}(B^T W^T \Phi WB + \bar{B}^T \bar{W}^T \Phi WB + B^T W^T \Phi \bar{W}\bar{B} + \bar{B}^T \bar{W}^T \Phi \bar{W}\bar{B}) \\ &= \mathrm{Tr}(B^T W^T W \Lambda B + \bar{B}^T \bar{W}^T W \Lambda B + B^T W^T \bar{W}\bar{\Lambda}\bar{B} + \bar{B}^T \bar{W}^T \bar{W}\bar{\Lambda}\bar{B}) \\ &= \mathrm{Tr}(B^T \Lambda B + 0 + 0 + \bar{B}^T \bar{\Lambda}\bar{B}) \\ &= \mathrm{Tr}(B^T \Lambda B + \bar{B}^T \bar{\Lambda}\bar{B}). \end{aligned}$$

By replacing $Z$ from $\mathcal{T}_3$ from Eq. (76), we get

$$
\begin{aligned}
-\operatorname{Tr}(Z^T Z \Lambda) &= -\operatorname{Tr}((WB + \bar{W}\bar{B})^T (WB + \bar{W}\bar{B})\Lambda) \\
&= -\operatorname{Tr}(B^T W^T W B \Lambda + \bar{B}^T \bar{W}^T W B \Lambda + B^T W^T \bar{W} \bar{B} \Lambda + \bar{B}^T \bar{W}^T \bar{W} \bar{B} \Lambda) \\
&= -\operatorname{Tr}(B^T B \Lambda + 0 + 0 + \bar{B}^T \bar{B} \Lambda) \\
&= -\operatorname{Tr}(B \Lambda B^T + \bar{B} \Lambda \bar{B}^T).
\end{aligned}
$$

The inequality that satisfies the 2nd order condition can now be written as

$$
\operatorname{Tr}(B^T \Lambda B) + \operatorname{Tr}(\bar{B}^T \bar{\Lambda} \bar{B}) - \operatorname{Tr}(B \Lambda B^T) - \operatorname{Tr}(\bar{B} \Lambda \bar{B}^T) + \mathcal{T}_1 \geq 0. \tag{77}
$$

Since $B$ is an antisymmetric matrix, $B^T = -B$, and therefore $\operatorname{Tr}(B \Lambda B^T) = \operatorname{Tr}(B^T \Lambda B)$. From this Eq. (77) can be rewritten as

$$
\operatorname{Tr}(B^T \Lambda B) - \operatorname{Tr}(B^T \Lambda B) + \operatorname{Tr}(\bar{B}^T \bar{\Lambda} \bar{B}) - \operatorname{Tr}(\bar{B} \Lambda \bar{B}^T) + \mathcal{T}_1 \geq 0. \tag{78}
$$

With the first two terms canceling each other out, the inequality can be rewritten as

$$
\operatorname{Tr}(\bar{B}^T \bar{\Lambda} \bar{B}) - \operatorname{Tr}(\bar{B} \Lambda \bar{B}^T) \geq -\mathcal{T}_1. \tag{79}
$$

With this inequality, the terms can be further bounded by

$$
\operatorname{Tr}(\bar{B}^T \bar{\Lambda} \bar{B}) \geq \min_i \bar{\Lambda}_i \operatorname{Tr}(\bar{B} \bar{B}^T)
$$

$$
\operatorname{Tr}(\bar{B} \Lambda \bar{B}^T) \leq \max_j \Lambda_j \operatorname{Tr}(\bar{B}^T \bar{B})
$$

Noting that since $\operatorname{Tr}(\bar{B} \bar{B}^T) = \operatorname{Tr}(\bar{B}^T \bar{B})$, we treat it as a constant value of $\alpha$. With this, the inequality can be rewritten as

$$
\left( \min_i \bar{\Lambda}_i - \max_j \Lambda_j \right) \geq -\frac{1}{\alpha} \mathcal{T}_1.
$$

Here, since $-\frac{1}{\alpha}\mathcal{T}_1$ is simply a constant, we denote it as $\mathcal{C}$ to yield the final conclusion that

$$
\left( \min_i \bar{\Lambda}_i - \max_j \Lambda_j \right) \geq \mathcal{C}. \tag{80}
$$

Eq. (80) concludes that to satisfy the 2nd order condition, the eigengap must be greater than $\mathcal{C}$. Therefore, given the choice of $q$ eigenvectors, the eigengap is maximized when the eigenvectors associated with the $q$ smallest eigenvalues are chosen as $W$. $\qquad \square$

We note that it is customary for machine learning algorithms to look for the most dominant eigenvectors, crucially, many KDR algorithms follow this standard. Therefore, to maintain consistency, the $\Phi$ defined within the paper is actually the negative $\Phi$ from the proof. By flipping the sign, the eigenvectors associated with the smallest eigenvalues is now the most dominant eigenvectors. Hence, $\Phi$ within the paper is defined as

$$
\Phi = \frac{1}{2} \sum_{ij} \Gamma_{ij} f'(\mathbf{a}^T WW^T \mathbf{b}) A_{i,j}. \tag{81}
$$

## Appendix E   Computing the Hessian for the Taylor Series

First we compute the gradient and the Hessian for $\beta(W)$ where

$$
\beta(W) = a^T WW^T b, \tag{82}
$$

$$
\beta(W) = \operatorname{Tr}(W^T b a^T W), \tag{83}
$$

$$
\nabla_W \beta(W) = [ba^T + ab^T]W, \tag{84}
$$

$$
\nabla_{W,W} \beta(W) = [ba^T + ab^T], \tag{85}
$$

$$
\nabla_{W,W} \beta(W = 0) = [ba^T + ab^T]. \tag{86}
$$

$$
\tag{87}
$$

Next, we compute the gradient and Hessian for $f(\beta(W))$ where

$$f(\beta(W)) = f(a^T W W^T b), \tag{88}$$

$$f(\beta(W)) = f(\text{Tr}(W^T b a^T W)), \tag{89}$$

$$f'(\beta(W)) = \nabla_\beta f(\beta(W))[ba^T + ab^T]W = \nabla_\beta f(\beta(W))\nabla_W \beta(W) \tag{90}$$

$$f''(\beta(W) = \nabla_{\beta,\beta} f(\beta(W))[ba^T + ab^T]W(...) + \nabla_\beta f(\beta(W))[ba^T + ab^T] \tag{91}$$

$$f''(\beta(W = 0)) = 0 + \nabla_\beta f(\beta(W))\nabla_{W,W} \beta(W = 0) \tag{92}$$

$$f''(\beta(W = 0)) = \nabla_\beta f(\beta(W))\nabla_{W,W} \beta(W = 0) \tag{93}$$

$$f''(0) = \mu A_{i,j}. \tag{94}$$

Using Taylor Series the gradient of the Lagrangian is approximately

$$\nabla_W \mathcal{L} \approx -\sum_{i,j} \Gamma_{i,j} f''(0) W - 2W\Lambda, \tag{95}$$

$$\nabla_W \mathcal{L} \approx -\mu \sum_{i,j} \Gamma_{i,j} A_{i,j} W - 2W\Lambda. \tag{96}$$

Setting the gradient of the Lagrangian to 0 and combining the constant 2 to $\mu$, it yields the relationship

$$\left[ -\mu \sum_{i,j} \Gamma_{i,j} A_{i,j} \right] W = W\Lambda. \tag{97}$$

Here we note that $\mu$ is a constant. Therefore, only the sign will affect the eigenvector selection. With this, it yields

$$\left[ -\text{sign}(\mu) \sum_{i,j} \Gamma_{i,j} A_{i,j} \right] W = W\Lambda. \tag{98}$$

With this, the terms within the bracket become the initial $\Phi_0$ as

$$\Phi_0 W = W\Lambda. \tag{99}$$

## Appendix F    Derivation for Approximated $\Phi$

We first convert the optimization problem into a standard minimization form where we solve

$$\min_W -\text{Tr}(\Gamma K_{XW}) \quad \text{s.t. } W^T W = I. \tag{100}$$

Since the objective Lagrangian is non-convex, a solution can be achieved faster and more accurately if the algorithm is initialized at an intelligent starting point. Ideally, we wish to have a closed-form solution that yields the global optimal without any iterations. However, this is not possible since $\Phi$ is a function of $W$. ISM circumvents this problem by approximating the kernel using Taylor Series up to the 2nd order while expanding around 0. This approximation has the benefit of removing the dependency of $W$ for $\Phi$, therefore, a global minimum can be achieved using the approximated kernel. The ISM algorithm uses the global minimum found from the approximated kernel as the initialization point. Here, we provide a generalized derivation for the ISM kernel functions that are twice differentiable. First, we note that the 2nd order Taylor expansion for $f(\beta(W))$ around 0 is $f(\beta(W)) \approx f(0) + \frac{1}{2!}\text{Tr}(W^T f''(0)W)$, where the 1st order expansion around 0 is equal to 0. Therefore, the ISM Lagrangian can be approximated with

$$\mathcal{L} = -\sum_{i,j} \Gamma_{i,j} \left[ f(0) + \frac{1}{2!}\text{Tr}(W^T f''(0)W) \right] - \text{Tr}(\Lambda(W^T W - I)), \tag{101}$$

where the gradient of the Lagrangian is

$$\nabla_W \mathcal{L} = -\sum_{i,j} \Gamma_{i,j} f''(0) W - 2W\Lambda. \tag{102}$$

Next, we look at the kernel function $f(\beta(W))$ more closely. The Hessian is computed as

$$f'(\beta(W)) = \nabla_\beta f(\beta(W))\nabla_W \beta(W), \tag{103}$$

$$f''(\beta(W=0)) = \nabla_\beta f(\beta(0))\nabla_{W,W}\beta(0). \tag{104}$$

Since we skipped several steps for the computation of the Hessian, refer to Appendix E for more detail. Because $\nabla_\beta f(\beta(0))$ is just a constant, we can bundle all constants into this term and refer to it as $\mu$. Since $\nabla_{W,W}\beta(0) = A_{i,j}$, the Hessian is simply $\mu A_{i,j}$ regardless of the kernel. By combining constants setting the gradient of Eq. (102) to 0, we get the expression

$$\left[ -\operatorname{sign}(\mu)\sum_{i,j}\Gamma_{i,j}A_{i,j} \right] W = W\Lambda, \tag{105}$$

where if we let $\Phi = -\operatorname{sign}(\mu)\sum_{i,j}\Gamma_{i,j}A_{i,j}$, we get a $\Phi$ that is not dependent on $W$. Therefore, a closed-form global minimum of the second-order approximation can be achieved. It should be noted that while the magnitude of $\mu$ can be ignored, the sign of $\mu$ cannot be neglected since it flips the sign of the eigenvalues of $\Psi$. Following Eq. (105), the initial $\Phi_0$ for each kernel is shown in Table 1. We also provide detailed proofs for each kernel in Appendix B.

It is important to note that based on proof of Theorem 1 in Appendix D, the $\Phi$ as defined from Eq. (105) requires the optimal $W$ to be the eigenvectors of $\Phi$ that is associated with the smallest eigenvalues. This is equivalent to the most dominant eigenvectors of negative $\Phi$. To maintain consistency, the $\Phi$ defined with the paper is the negative $\Phi_0$ from this derivation, and therefore the $\Phi_0$ defined within the paper is

$$\Phi = \operatorname{sign}(\mu)\sum_{i,j}\Gamma_{i,j}A_{i,j}. \tag{106}$$

## Appendix G   Theorem 12.5

**Lemma 3** (Nocedal,Wright, Theorem 12.5 [32]). *(2nd Order Necessary Conditions) Consider the optimization problem:* $\min_{W:h(W)=0} f(W)$, *where* $f : \mathbb{R}^{d\times q} \to \mathbb{R}$ *and* $h : \mathcal{R}^{d\times q} \to \mathbb{R}^{q\times q}$ *are twice continuously differentiable. Let* $\mathcal{L}$ *be the Lagrangian and* $h(W)$ *its equality constraint. Then, a local minimum must satisfy the following conditions:*

$$\nabla_W \mathcal{L}(W^*, \Lambda^*) = 0, \tag{107a}$$

$$\nabla_\Lambda \mathcal{L}(W^*, \Lambda^*) = 0, \tag{107b}$$

$$\operatorname{Tr}(Z^T \nabla^2_{WW}\mathcal{L}(W^*, \Lambda^*)Z) \geq 0$$
$$\text{for all } Z \neq 0, \text{with } \nabla h(W^*)^T Z = 0. \tag{107c}$$

## Appendix H   Derivation for $\sum_{i,j}\Psi_{i,j}A_{i,j}$ if $A_{i,j} = x_i x_j^T + x_j x_i^T$

Since $\Psi$ is a symmetric matrix and $A_{i,j} = (x_i x_j^T + x_j x_i^T)$, we first note that while $x_i x_j^T \neq x_j x_i^T$, it still hold that

$$\sum_{i,j}\Psi_{i,j}x_i x_j^T = \sum_{i,j}\Psi_{i,j}x_j x_i^T. \tag{108}$$

Therefore, we can rewrite the expression into

$$\sum_{i,j}\Psi_{i,j}A_{i,j} = 2\sum_{i,j}^{n}\Psi_{i,j}x_i x_j^T.$$

If we expand the summation for $i = 1$, we get

$$
\begin{aligned}
\left[\Psi_{1,1}x_1 x_1^T + \ldots + \Psi_{1,n}x_1 x_n^T\right] &= x_1\left[\Psi_{1,1}x_1^T + \ldots + \Psi_{1,n}x_n^T\right] \\
&= x_1\left[\begin{bmatrix} x_1 & \ldots & x_n \end{bmatrix}\begin{bmatrix} \Psi_{1,1} \\ . \\ \Psi_{1,n} \end{bmatrix}\right]^T \\
&= x_1\left[\begin{bmatrix} \Psi_{1,1} & \ldots & \Psi_{1,n} \end{bmatrix}\begin{bmatrix} x_1^T \\ . \\ x_n^T \end{bmatrix}\right].
\end{aligned}
$$

Now if we sum up all $i$, we get

$$
\begin{aligned}
\Psi_{i,j}x_ix_j^T &= x_1\left[\begin{bmatrix} \Psi_{1,1} & \cdots & \Psi_{1,n} \end{bmatrix}\begin{bmatrix} x_1^T \\ \cdot \\ x_n^T \end{bmatrix}\right] + \ldots + x_n\left[\begin{bmatrix} \Psi_{n,1} & \cdots & \Psi_{n,n} \end{bmatrix}\begin{bmatrix} x_1^T \\ \cdot \\ x_n^T \end{bmatrix}\right], \\
&= \left[x_1\begin{bmatrix} \Psi_{1,1} & \cdots & \Psi_{1,n} \end{bmatrix} + \ldots + x_n\begin{bmatrix} \Psi_{n,1} & \cdots & \Psi_{n,n} \end{bmatrix}\right]\begin{bmatrix} x_1^T \\ \cdot \\ x_n^T \end{bmatrix}, \\
&= \left[\begin{bmatrix} x_1 & \cdots & x_n \end{bmatrix}\begin{bmatrix} \Psi_{1,1} \\ \cdot \\ \Psi_{n,1} \end{bmatrix} + \ldots + \begin{bmatrix} x_1 & \cdots & x_n \end{bmatrix}\begin{bmatrix} \Psi_{1,n} \\ \cdot \\ \Psi_{n,n} \end{bmatrix}\right]\begin{bmatrix} x_1^T \\ \cdot \\ x_n^T \end{bmatrix}, \\
&= \begin{bmatrix} x_1 & \cdots & x_n \end{bmatrix}\left[\begin{bmatrix} \Psi_{1,1} \\ \cdot \\ \Psi_{n,1} \end{bmatrix} \cdots \begin{bmatrix} \Psi_{1,n} \\ \cdot \\ \Psi_{n,n} \end{bmatrix}\right]\begin{bmatrix} x_1^T \\ \cdot \\ x_n^T \end{bmatrix}.
\end{aligned}
$$

Given that $X = \begin{bmatrix} x_1 & \cdots & x_n \end{bmatrix}^T$, the final expression becomes.

$$
2\sum_{i,j}^{n}\Psi_{i,j}x_ix_j^T = 2X^T\Psi X.
$$

# Appendix I   Derivation for $\sum_{i,j}\Psi_{i,j}A_{i,j}$ if
$$A_{i,j} = (x_i - x_j)(x_i - x_j)^T + (x_i - x_j)(x_i - x_j)^T$$

Since $\Psi$ is a symmetric matrix, and $A_{i,j} = (x_i - x_j)(x_i - x_j)^T + (x_i - x_j)(x_i - x_j)^T = 2(x_i - x_j)(x_i - x_j)^T$, we can rewrite the expression into

$$
\begin{aligned}
\sum_{i,j}\Psi_{i,j}A_{i,j} &= 2\sum_{i,j}\Psi_{i,j}(x_i - x_j)(x_i - x_j)^T \\
&= 2\sum_{i,j}\Psi_{i,j}(x_ix_i^T - x_jx_i^T - x_ix_j^T + x_jx_j^T) \\
&= 4\sum_{i,j}\Psi_{i,j}(x_ix_i^T - x_jx_i^T) \\
&= \left[4\sum_{i,j}\Psi_{i,j}(x_ix_i^T)\right] - \left[4\sum_{i,j}\Psi_{i,j}(x_jx_i^T)\right].
\end{aligned}
$$

If we expand the 1st term where $i = 1$, we get

$$
\sum_{i=1,j}^{n}\Psi_{1,j}(x_1x_1^T) = \Psi_{1,1}(x_1x_1^T) + \ldots + \Psi_{1,n}(x_1x_1^T) = \left[\sum_{i=1,j}^{n}\Psi_{1,j}\right]x_1x_1^T.
$$

From here, we notice that $\left[\sum_{i=1,j}^{n}\Psi_{1,j}\right]$ is the degree $d_{i=1}$ of $\Psi_{i=1}$. Therefore, if we sum up all $i$ values we get

$$
\sum_{i,j}\Psi_{i,j}(x_ix_i^T) = d_1x_1x_1^T + \ldots + d_nx_nx_n^T.
$$

If we let $D_\Psi$ be the degree matrix of $\Psi$, then this expression becomes

$$
4\sum_{i,j}\Psi_{i,j}(x_ix_i^T) = 4X^TD_\Psi X.
$$

Since Appendix H has already proven the 2nd term, together we get

$$
4\sum_{i,j}\Psi_{i,j}(x_ix_i^T) - 4\sum_{i,j}\Psi_{i,j}(x_jx_i^T) = 4X^TD_\Psi X - 4X^T\Psi X = 4X^T[D_\Psi - \Psi]X.
$$

## Appendix J   Dataset Details

**Wine.**   This dataset has 13 features and 178 samples. The features are continuous and heavily unbalanced in magnitude. During the experiments, the dimension is reduced down to 3 prior to performing supervised or unsupervised tasks. The dataset can be downloaded at `https://archive.ics.uci.edu/ml/datasets/wine`.

**Cancer.**   This dataset has 9 features and 683 samples. The features are discrete and unbalanced in magnitude. During the experiments, the dimension is reduced down to 2 prior to performing supervised or unsupervised tasks. The dataset can be downloaded at `https://archive.ics.uci.edu/ml/datasets/Breast+Cancer+Wisconsin+(Diagnostic)`.

**Face.**   This dataset consists of images of 20 people in various poses. The 624 images are vectorized into 960 features. During the experiments, the dimension is reduced down to 20 prior to performing supervised or unsupervised tasks. This dataset is commonly used for alternative clustering since it can be clustered by the identity or the pose of the individuals. The dataset can be downloaded at `https://archive.ics.uci.edu/ml/datasets/CMU+Face+Images`.

**MNIST.**   This dataset consists of 10,000 images of 10 characters in various orientations. The images are vectorized into 785 features. During the experiments, the dimension is reduced down to 10 prior to performing supervised or unsupervised tasks. The original MNIST dataset consists of 60,000 training samples and 10,000 test samples. We have decided to use the 10,000 test samples as our dataset. Since ISM have a memory complexity of $O(n^2)$, storing matrix size of 60,000 $\times$ 60,000 was beyond our computer's capability. We are actively conducting research into using the concept of coresets to alleviate the memory bottleneck. The dataset can be downloaded at `http://yann.lecun.com/exdb/mnist/`

**Flower.**   The Flower image is a dataset that allows for alternative ways to perform image segmentation. It is an image of 350x256 pixel. The RGB values of each pixel is taken as a single sample, with repeated samples removed. This results in a dataset of 256 samples and 3 features. The image is segmented into group of 2, represented by black and white. The dataset can be downloaded at `http://en.tessellations-nicolas.com/`

## Appendix K   Proof of Reformulating Eq. (1) into Quadratic Optimization

Given

$$\min_W \operatorname{Tr}(W^T \Phi W) \quad \text{s.t.} \quad W^T W = I. \tag{109}$$

Here we proof that the local minimum for Eq. (109) is equivalent to a local minimum for Eq. (1). From Theorem 1, we establish that the $q$ minimizing eigenvectors of $\Phi \in \mathcal{R}^{d \times d}$ is a local minimum of Eq. (1). Therefore, the strategy of this proof is to show that the optimal solution for Eq. (109) is also the minimizing eigenvectors of $\Phi$.

*Proof.*   Given Eq. (109), the Lagrangian of the objective is

$$\mathcal{L}(W) = \operatorname{Tr}(W^T \Phi W) - \operatorname{Tr}\left[\Lambda(W^T W - I)\right]. \tag{110}$$

Therefore, given a symmetric $\Phi$, the gradient of the Lagrangian becomes

$$\nabla_W \mathcal{L}(W) = 2\Phi W - 2W\Lambda. \tag{111}$$

Here, by setting the gradient to 0, we arrive to the definition of eigenvector where

$$\Phi W = W\Lambda, \tag{112}$$

thereby proving that the eigenvector of $\Phi$ is also a stationary point for Eq. (109). The proof ends here if $W \in \mathcal{R}^{d \times d}$, however, if $W \in \mathcal{R}^{d \times q}$ where $q < d$, then we must also determine the appropriate $q$ eigenvectors to minimize the objective. Given $\bar{W} \in \mathcal{R}^{d \times d}$ as the full set eigenvectors, we replace $W$ from Eq. (110) with $\bar{W}$ to get

$$\mathcal{L}(\bar{W}) = \operatorname{Tr}(\bar{W}^T \Phi \bar{W}) - \operatorname{Tr}\left[\Lambda(\bar{W}^T \bar{W} - I)\right]. \tag{113}$$

Since $\bar{W}^T\bar{W} = I$ and $\Phi\bar{W} = W\Lambda$, we substitute these terms into Eq. (113) to get

$$\mathcal{L}(\bar{W}) = \text{Tr}(\bar{W}^T\bar{W}\Lambda). \tag{114}$$

If we let $w_1, w_2, ..., w_d$ be the set of individual eigenvectors of $\Phi$ within $\bar{W}$ and $\lambda_1, \lambda_2, ..., \lambda_d$ be their corresponding eigenvalues, then Eq. (114) can be rewritten as

$$\mathcal{L}(\bar{W}) = \lambda_1 w_1^T w_1 + \lambda_2 w_2^T w_2 + ... + \lambda_n w_d^T w_d. \tag{115}$$

Since the inner product of any eigenvector with itself $(w_i^T w_i)$ is always equal to 1, the Lagrangian becomes the summation of its eigenvalues where

$$\mathcal{L}(\bar{W}) = \lambda_1 + \lambda_2 + ... + \lambda_n. \tag{116}$$

Therefore, the selection of a subset of eigenvectors is equivalent to keeping a subset of eigenvalues while setting the rest to 0 in Eq. (116). To minimize the Lagrangian, therefore, implies that the eigenvectors corresponding to the smallest eigenvalues should be chosen. Here, we have proven that the minimizing eigenvectors of $\Phi$ is a local minimum for both Eq. (1) and (109). □

## Appendix L  NMI Calculation

If we let $U$ and $L$ be two clustering assignments, NMI can be calculated with

$$NMI(L,U) = \frac{I(L,U)}{\sqrt{H(L)H(U)}}, \tag{117}$$

where $I(L,U)$ is the mutual information between $L$ and $U$, and $H(L)$ and $H(U)$ are the entropies of $L$ and $U$ respectively.

## Appendix M  Proof for Corollary 1

*Proof.* The optimization of Eq. (1) using a conic combination of $m$ kernels becomes

$$\min_W - \text{Tr}\left(\Gamma[\mu_1 K_1 + \mu_2 K_2 + ... + \mu_m K_m]\right) \quad \text{s.t. } W^T W = I. \tag{118}$$

The trace term can be separated into dividual terms where

$$\min_W - \text{Tr}(\mu_1\Gamma K_1) - \text{Tr}(\mu_2\Gamma K_2) - ... - \text{Tr}(\mu_m\Gamma K_m) \quad \text{s.t. } W^T W = I. \tag{119}$$

Therefore, the Lagrangian can be written as

$$\mathcal{L} = - \text{Tr}(\mu_1\Gamma K_1) - \text{Tr}(\mu_2\Gamma K_2) - ... - \text{Tr}(\mu_m\Gamma K_m) - mTr(\Lambda[W^T W - I]). \tag{120}$$

From Lemma 1, we have shown that the gradient of the Lagrangian becomes

$$\nabla_W\mathcal{L} = [-\mu_1\Phi_1 - \mu_2\Phi_2 - ... - \mu_m\Phi_m]W - mW\Lambda, \tag{121}$$

where each $\Phi_i$ is the $\Phi$ matrix corresponding to each kernel. Setting the gradient to 0, it yields the relationship

$$\frac{1}{m}[-\mu_1\Phi_1 - \mu_2\Phi_2 - ... - \mu_m\Phi_m]W = W\Lambda, \tag{122}$$

Therefore, optimizing a conic combination of kernels for Eq. (1) is equivalent to using a conic combination of the corresponding $\Phi$s with the same coefficients. □

## Appendix N  An Overview on HSIC

Proposed by Gretton et al. [18], the Hilbert Schmidt Independence Criterion (HSIC) is a statistical dependence measure between two random variables. HSIC is similar to mutual information (MI) because given two random variables $X$ and $Y$, they both measure the distance between the joint

distribution $P_{X,Y}$ and the product of their individual distributions $P_X P_Y$. While MI uses KL-divergence to measure this distance, HSIC uses Maximum Mean Discrepancy [43]. Therefore, when HSIC is zero, or $P_{X,Y} = P_X P_Y$, it implies independence between $X$ and $Y$. Similar to MI, HSIC score increases as $P_{X,Y}$ and $P_X P_Y$ move away from each other, thereby also increasing their dependence. Although HSIC is similar to MI in its ability to measure dependence, it is easier to compute as it removes the need to estimate the joint distribution.

Formally, given a set of $N$ i.i.d. samples $\{(x_1, y_1), ..., (x_N, y_N)\}$ drawn from a joint distribution $P_{X,Y}$. Let $X \in \mathbb{R}^{N \times d}$ and $Y \in \mathbb{R}^{N \times c}$ be the corresponding sample matrices where $d$ and $c$ denote the dimensions of the datasets. We denote by $K_X, K_Y \in \mathbb{R}^{N \times N}$ the kernel matrices with entries $K_{X_{i,j}} = k_X(x_i, x_j)$ and $K_{Y_{i,j}} = k_Y(y_i, y_j)$, where $k_X : \mathbb{R}^d \times \mathbb{R}^d \to \mathbb{R}$ and $k_Y : \mathbb{R}^c \times \mathbb{R}^c \to \mathbb{R}$ represent kernel functions. Furthermore, let $H$ be a centering matrix defined as $H = I_n - \frac{1}{n} \mathbf{1}_n \mathbf{1}_n^T$ where $\mathbf{1}_n$ is a column vector of ones. HSIC is computed empirically with

$$\mathbb{H}(X, Y) = \frac{1}{(n-1)^2} \operatorname{Tr}(K_X H K_Y H). \tag{123}$$

## Appendix O    Proof for Proposition 1

*Proof.* For a kernel to belong to the ISM family, it must satisfy the following 3 conditions.

- The kernel function must be twice differentiable.

- The kernel function can be written in terms of $f(\beta)$.

- The kernel matrix from $f(\beta)$ must be symmetric positive semi-definite.

To satisfy the 1st condition, given a kernel $K$ that is a conic combination of $n$ ISM kernels where

$$K = \mu_1 K_1 + \mu_2 K_2 + ... + \mu_n K_n. \tag{124}$$

Since each kernel $K_i$ is twice differentiable, the conic combination is still twice differentiable. Therefore, $K$ is a twice differentiable function.

To satisfy the 2nd condition, given a kernel $K$ from Eq. (124) where each kernel is from the ISM family, the trivial case of when $\beta = a(x_i, x_j) W W^t b(x_i, x_j)$ is defined identically between kernels:

$$K = \mu_1 f_1(\beta) + \mu_2 f_2(\beta) + ... + \mu_n f_n(\beta). \tag{125}$$

From Eq. (125), it is obvious that $K$ itself can also be written in terms of $\beta$. However, in the cases where the functions $a(x_i, x_j)$ and $b(x_i, x_j)$ are defined differently, $\beta$ must be defined differently. Here, we define the following

$$a(x_i, x_j) = \mathbf{Diag}([a_1(x_i, x_j), a_2(x_i, x_j), .., a_n(x_i, x_j)]) \tag{126}$$

$$b(x_i, x_j) = \mathbf{Diag}([b_1(x_i, x_j), b_2(x_i, x_j), .., b_n(x_i, x_j)]) \tag{127}$$

$$W = \mathbf{Diag}([W, W, ..., W]) \tag{128}$$

$$W^T = \mathbf{Diag}([W^T, W^T, ..., W^T]) \tag{129}$$

$$\beta = \mathbf{Diag}([a_1^T W W^T b_1, a_2^T W W^T b_2, ..., a_n^T W W^T b_n]) \tag{130}$$

where **Diag** puts the element of the vector on the diagonal of a matrix with both the upper and lower triangle as 0s. Given $\beta$ is a matrix, each kernel function can always multiply $\beta$ by a one-hot vector on both sides to choose the appropriate sub-$\beta$ value. Therefore, the joint kernel $K$ can always be written in terms of $\beta$.

For the 3rd condition, we know that conic combinations of symmetric positive semi-definite matrices are still symmetric positive semi-definite. □

## Appendix P    Proof of Theorem 2

*Proof.* The original ISM leverages Bolzano-Weierstrass theorem to prove that a sequence generated using the Gaussian kernel is bounded, therefore ISM has a convergent subsequence. Since the

generalized ISM extends the guarantee to other kernels, here we demonstrate that the extension of ISM to other kernels does not have any effect on the convergence guarantee.

ISM over arbitrary kernels solves an optimization problem over the Grassmannian manifold $G(n, d)$, as parametized by the subspace $WW^T$. The Grassmannian manifold is a quotient of the Stiefel Manifold $G(n, d) = V(n, d)/O(n)$. The Grassmann manifold inherits compactness and an induced metric from the Stiefel manifold. For metric spaces, compact and sequentially compact topological spaces are equivalent. Therefore, sequences $\{WW^T\}^k$ will have convergent subsequences. While $W$ may not converge (choice of frame), its subspace description will. Termination criteria in Algorithm 1 is independent of the frame $W$.

$\square$

## Appendix Q  Convergence Criteria

Since the objective is to discover a linear subspace, the rotation of the space does not affect the solution. Therefore, instead of constraining the solution on the Stiefel Manifold, the manifold can be relaxed to a Grassmann Manifold. This implies that Algorithm 1 can reach convergence as long as the columns space spanned by $W$ are identical. To identify the overlapping span of two spaces, we can append the two matrices into $\mathcal{W} = [W_k W_{k+1}]$ and observe the rank of $\mathcal{W}$. In theory, the rank should equal to $q$, however, a hard threshold on rank often suffers from numerical inaccuracies.

One approach is to study the principal angles ('angles between flats') between the subspaces spanned by $W_k$ and $W_{k+1}$. This is based on the observation that if the maximal principal angle $\theta_{\max} = 0$, then the two subspaces span the same space. The maximal principal angle between subspaces spanned by $W_k$ and $W_{k+1}$ can be found by computing $U\Sigma V^T = W_k^T W_{k+1}$ [44]. The cosines of the principal angles between $W_k$ and $W_{k+1}$ are the singular values of $\Sigma$, thus $\theta_{\max} = \cos^{-1}(\sigma_{\min})$. Computation of $\theta_{\max}$ requires two matrix multiplications to form $V\Sigma^2 V^T = (W_k^T W_{k+1})^T (W_k^T W_{k+1})$ and then a round of inverse iteration to find $\sigma_{\min}^2$. Although this approach confirms the convergence definitively, in practice, we avoid this extra computation by using the convergence of eigenvalues (of $\Phi$) between iterations as a surrogate. Since eigenvalues are already computed during the algorithm, no additional computations are required. Although tracking eigenvalue of $\Phi$ for convergence is vulnerable to false positive errors, in practice, it works consistently well. Therefore, we recommend to use the eigenvalues as a preliminary check before defaulting to principal angles.