[Reviews · NeurIPS 2019]

Reviewer 1



The listing of 14 references in the first sentence (without providing any information about them) is unnecessary. “For supervised dimension reduction, we perform SVM on XW using 10-fold cross validation after learning W” This is a clear problem in the experiments. The generalization of the DR method should be tested on out of sample data. Since the final mapping is a linear projection Z = X*W, comparisons should also include linear subspace learning methods. Comparisons should also include the use of kPCA (or approximation using Nystrom method) for the data projection.

Reviewer 2



The paper extended the theoretical guarantees of ISM to a family of kernels beyond the Gaussian kernel. The paper also showed that a conic combination of ISM family of kernels stays as a valid ISM kernel. Pro: 1. ISM can be utilized as an efficient solution to a variety of IKDR learning paradigms. 2. Experiments on both supervised and unsupervised learning paradigms and for a variety of kernels showcase the generality of ISM. 3. verify the improvement in speed offered by ISM as an optimization algorithm for solving IKDR compared to competing methods and with better or comparable accuracy. Cons, the novelty seems not high. 1. the method and algorithm seem straightforward. 2. experiments are more toy

Reviewer 3



Summary: [19] has proposed recently an efficient iterative spectral (eigendecomposition) method (ISM) for the non-convex interpretable kernel dimensionality reduction (IKDR) objective in the context of alternative clustering. It established theoretical guarantees of ISM for the Gaussian kernel. The paper extends the theoretical guarantees of ISM to a family of kernels [Definition 1]. Each kernel in the ISM family has an associated surrogate matrix \Phi and the optimal projection is formed by the most dominant eigenvectors of \Phi [Theorem 1 and 2]. They showed that any conic combination of the ISM kernels is still an ISM kernel [Proposition 1] and therefore ISM can be extend to conic combination of kernels. They further showed how a range of different learning paradigms (i.e. supervised, unsupervised, semi-supervised dimension reduction and alternative clustering) can all be formulated as a constraint optimisation problem (equation 1) which can be solved by ISM. Hence the realisation of ISM’s potential through the generalisation into a wider range of kernels impacts a wide range of applications. The performance of the ISM using a range of kernels from the ISM family is illustrated in 5 real data experiments under different learning paradigms. Quality: The theoretical claims are well supported by theoretical analysis. A variety of experimental results are presented showcasing the application of the ISM algorithm on a variety of learning paradigms using a variety of kernels. Clarity: Overall, the paper is written with clear structure. However, there are a few places that confuses me. - The paragraph starting line 153, the authors starts to introduce the matrices used and using the language from group theory I believe? Can the authors elaborate on the reasons and connections (if any)? - Can the authors provide some intuitions on the ISM kernels and the associated phi matrix? All the proofs seems to fall magically into place using this phi matrix, so in some sense this phi matrix capture all the informations about the corresponding kernel matrix? Is this correspondence unique? - The matrix A was first defined on p.16 in the proof of Lemma 1, however was first mentioned in Theorem 3. - For the notation in Definition 1 and later the proof of Lemma 1, I am a little confused by what a(x_i, x_j) is? A function with two arguments? Can it be any functions? (I notice the authors discussed when a = x_i, or when a = x_i - x_j, but when do we use which (some derivation of the phi matrix used the former while others used the later)? Some clear explanation is needed here.) Also, what does {\bf a } = a(x_i, x_j ) mean? - In Table 4, MNIST dataset the row with NMI, why does it suddenly change into percentage and why are both bolded? Also, for the wine example, why are NMI with 0.86 and 0.84 all bolded? Originality: The main contribution of the work is the extension of the theoretical guarantees of ISM [19] to a family of kernels and any conic combination of kernels from this family of kernels. While showing that different objective functions from a variety of IKDR problems can be formulated into the same form, the paper illustrates the wider applicability of the ISM algorithm. Though established some new/improved results on the existing datasets, I feel the main contributions of the paper are theoretical. Significance: The realisation of ISM’s potential through the generalisation into a wider range of kernels impacts a wide range of applications. We saw in the experiments, ISM can now be applied to solve a variety of IKDR problems with better computational efficiency comparing to other state of art manifold optimisations. Comparable and sometimes better performance are observed by utilising a different kernel than the Gaussian kernel. Practitioners are likely to use the ideas. ======================= Authors response read. I am happy with the response provided.

[Author Response · NeurIPS 2019]

Thank you for the reviews and suggestions; we will revise accordingly.

**Reviewer 1**
**On the 14 references:** Thank you for the suggestion, the period before the list was a typo and the list of references
were examples of dimension reduction algorithms supporting the 1st sentence. We will condense to fewer citations.

**On 10-Fold:** Thank you for pointing out this potential for misunderstanding. For each of the 10-fold experiments, we
trained $W$ and the SVM classifier only on the training set while reporting the result only on the test set. The test set was
never used during the training; training and testing data are strictly separated. We repeat this process for each fold of
cross-validation. We will make sure to clarify the training/testing process in the camera ready version.

**On additional experiments:** There are 4 central claims of the paper. First, ISM can be generalized to many learning
paradigms (supervised, semi-supervised, unsupervised). Second, ISM is significantly faster than existing optimization
algorithms for IKDR. Third, ISM can generalize to multiple kernels with the appropriate $\Phi$ matrix. Lastly, the ISM
family extends to conic combinations of kernels in the ISM family.

Like the first reviewer suggested, we conducted experiments with PCA and KPCA in a preliminary draft. Against PCA,
IKDR significantly reduced the error rate while maintaining similar execution time across all datasets. Against KPCA,
IKDR simultaneously reduced the error rate and the execution time. Although these results were interesting, when
trimming the submission version we decided to focus the experiments on supporting the 4 central claims of the paper.
We will add these results in our supplement on final revision.

**Reviewer 2**
**On the algorithm seem straightforward:** Our contribution focuses on the theoretical aspects of ISM for solving
IKDR for a general class of kernels and our experiments are designed to support the claims. The ISM algorithm (which
is an iterative eigendecomposition) itself is proposed previously by [19] and is not a part of our claim of contribution.
We found the ISM algorithm simple and elegant, but its algorithmic simplicity hides the rigorous analysis required to
guarantee its effectiveness. The sizable proofs provided in the appendix is a testament of the complexity involved to
further extend the theoretical guarantees of ISM to other kernels for solving highly non-convex IKDR problems.

In addition to the theoretical contributions, another novelty of this work is the discovery of the $\Phi$ matrices *unique* to
each ISM kernel. Indeed, every kernel within the ISM family possesses an alternative representation that is significantly
smaller and yet contains all the necessary information. For IKDR applications, we discovered a matrix $\Phi$ that can
replace kernels while significantly reducing its computational complexity. We consider discovering an alternative
efficient representation to kernels of great interest for the IKDR community.

**On experiments using toy data:** All of the 5 datasets are real and not synthetically generated. They were carefully
chosen for convenient comparison since many related work also used the same datasets. They were also carefully
chosen to span different data types. The Flower and Face datasets are real images commonly used for alternative
clustering. The cancer and wine datasets are used for many supervised and unsupervised domains while having both
discrete and continuous features respectively. The MNIST handwritten digit dataset is a standard benchmark researchers
use to compare classification performance. See citations 33-37.

**Reviewer 3**
**On clarifying line 153:** We apologize for the confusion. The language in this paragraph is commonly used in Spectral
Clustering from Graph Theory. For a given matrix $\Psi$, its degree matrix is $D_\Psi = \text{Diag}(\Psi 1_n)$ where $1_n$ is a vector of 1s
and Diag is a function that places the elements of a vector into the diagonal of a zero squared matrix. The Laplacian
matrix is $\mathcal{L} = D_\Psi - \Psi$. The connection to graph theory is that $\Psi$ is treated as a weighted adjacency matrix, and the
Laplacian $\mathcal{L}$ shows up in many resulting formulas. We'll provide additional information to clarify the language.

**On the intuition of the $\Phi$ matrix:** Observe from Table 3 that $\Phi$ can be expressed as $X^T \Omega X$, where $\Omega$ is a positive
semi-definite (PSD) matrix. Removing $\Omega$, $X^T X$ is simply the covariance matrix. Applying Cholesky decomposition
on $\Omega$ yields $(X^T L)(L^T X)$. Since the PSD $\Omega$ matrix is constructed from the kernel and label information, $\Omega$ is an
interpretable way to scale the covariance matrix using the kernel and label information. We'll add this intuition.

**On matrix $A$:** Thank you for finding this oversight, we will include its definition within the paper.

**On Definition 1:** Given data $x_i \in \mathbb{R}^d$, $a(x_i, x_j)$ is a function defined as $a : \mathbb{R}^d \times \mathbb{R}^d \to \mathbb{R}^d$. As a short hand, $\mathbf{a}$ is the
image of $a$. From our analysis, $a$ can be any function that allows a kernel function to be written in terms of $f(\beta)$ while
remaining a valid kernel. The definition of $a(x_i, x_j)$ depends on the kernel shown in Table 1. $b$ and $\mathbf{b}$ are defined the
same way. We thank the reviewer for this point of confusion and we will address these definitions in the final release.

**On Table 4:** Thank you for noticing the minor typo. Instead of 47%, it is supposed to be 0.47. We will make this
update accordingly. For the wine dataset, the bold text highlights the best performing results. Since 0.86 and 0.84 are
all the best results from Gaussian and Polynomial, they were all highlighted.

[Meta-Review · NeurIPS 2019]

This paper proposes an efficient computation of kernel dimension reduction, extending the previous work [19] to much more general class of kernels. We think this extension has an impact both in theory and practice from the general applicability of the framework.